# High mid-Holocene accumulation rates over West Antarctica inferred from a pervasive ice-penetrating radar reflector

Julien A. Bodart[1], Robert G. Bingham[1], Duncan A. Young[2], Joseph A. MacGregor[3], David W. Ashmore[4,5], Enrica Quartini[2,6], Andrew S. Hein[1], David G. Vaughan[7†], and Donald D. Blankenship[2]

[1]School of GeoSciences, University of Edinburgh, Edinburgh, UK
[2]Institute for Geophysics, University of Texas at Austin, Austin, Texas, USA
[3]Cryospheric Sciences Laboratory, NASA Goddard Space Flight Center, Greenbelt, Maryland, USA
[4]School of Environmental Sciences, University of Liverpool, Liverpool, UK
[5]Met Office, Exeter, UK
[6]Department of Astronomy, Cornell University, Ithaca, New York, USA
[7]British Antarctic Survey, Cambridge, UK
† Deceased

*Correspondence to*: Julien A. Bodart (julien.bodart@ed.ac.uk)

**Key points**

- We estimate mean accumulation rates for the past ~4700 years across the Pine Island, Thwaites, and Institute and Möller ice-stream catchments in West Antarctica using a ubiquitous, ice-core dated internal radar reflection
- Accumulation rates were 18% higher during the mid-Holocene compared to modern rates over the Amundsen-Weddell-Ross divide
- Spin-up of regional and continental ice-sheet models should include time-varying changes in Holocene accumulation rates from the WAIS Divide Ice Core to generate more realistic grounding-line evolution and past sea level rise contribution across this region

**Abstract**

Understanding the past and future evolution of the Antarctic Ice Sheet is challenged by the availability and quality of observed palaeo-boundary conditions. Numerical ice-sheet models often rely on these palaeo-boundary conditions to guide and evaluate their models' predictions of sea-level rise, with varying levels of confidence due to the sparsity of existing data across the ice sheet. A key data source for large-scale reconstruction of past ice-sheet processes are Internal Reflecting Horizons (IRHs) detected by Radio-Echo Sounding (RES). When IRHs are isochronal and dated at ice cores, they can be used to determine palaeo-accumulation rates and patterns on large spatial scales. Using a spatially extensive IRH over Pine Island Glacier, Thwaites Glacier, and Institute and Möller Ice Streams (covering a total of 610 000 km$^2$ or 30% of the WAIS), and a local layer approximation model, we infer mid-Holocene accumulation rates over the slow-flowing parts of these catchments for the past ~4700 years. By comparing our results with modern climate reanalysis models (1979 – 2019) and observational syntheses (1651 – 2010), we estimate that accumulation rates over the Amundsen-Weddell-Ross divide were on average 18% higher during the mid-Holocene than modern rates. However, no significant spatial changes in the accumulation pattern were observed. The higher mid-Holocene accumulation-rate estimates match previous palaeo-accumulation estimates from ice-core records and targeted RES surveys over the ice divide, and they also coincide with periods of grounding-line readvance during the Holocene over the Weddell and Ross Sea sectors. We find that our spatially-extensive, mid-Holocene-to-present accumulation estimates are consistent with a sustained late-Holocene period of higher accumulation rates occurring over millennia reconstructed from the WAIS Divide Ice Core, thus indicating that this ice core is spatially representative of the wider West Antarctic region. We conclude that future regional and continental ice-sheet modelling studies should base their climatic forcings on time-varying accumulation rates from the WAIS Divide Ice Core through the Holocene to generate more realistic predictions of the West Antarctic Ice Sheet's past contribution to sea-level rise.

**Key words:** West Antarctica, Internal Reflecting Horizons, Accumulation, Holocene, Ice-Penetrating Radars, Ice-Core, Pine Island Glacier, Thwaites Glacier.

## 1. Introduction

Improving our knowledge of past climatic changes over the Antarctic Ice Sheet is required if we are to understand its present evolution and model its future under increasingly rapid climatic changes (IPCC, 2021). Most studies of past ice-sheet behaviour over Antarctica have focused on modelling changes in ice volume and grounding-line retreat following the Last Glacial Maximum (LGM, ~20 ka Before Present, BP) (Denton and Hughes, 2002; Golledge et al., 2012; 2013; Hillenbrand et al., 2013; 2014; Le Brocq et al., 2011; Kingslake et al., 2018); however, less attention has been paid to ice-sheet evolution during the Holocene (~11.7 ka BP to present). Recent evidence suggests that parts of the grounding line of West Antarctica may have retreated several hundred kilometres inland from its current position at ~10 ka and subsequently readvanced to reach its modern position sometime during the Holocene, due to isostatic rebound and climate-induced changes, particularly over the Weddell Sea and western Ross Sea sectors (Siegert et al., 2013; Bradley et al., 2015; Kingslake et al., 2018; Wearing and Kingslake, 2019; Venturelli et al., 2020; Neuhaus et al., 2021; Johnson et al., 2022). However, the atmospheric and ice-dynamical conditions farther inland, which could also have induced grounding-line migration, remain poorly constrained. An early investigation by Whillans (1976) using radar data near Byrd Ice Core indicated stability during the Late Pleistocene and Holocene epochs. Records of temperature and precipitation from the WAIS Divide Ice Core (hereafter abbreviated as WD14; Fig. 1) in the central West Antarctic Ice Sheet (WAIS) suggest higher accumulation rates during the Holocene than at present (Fudge et al., 2016), a trend that is also observed across small parts of the Amundsen-Weddell-Ross divide (Fig. 1) near the WAIS Divide Ice Core (hereafter referred to as WD14; Fig. 1) where isolated Radio-Echo Sounding (RES) surveys have shown 15-30% higher accumulation rates during the mid-Holocene compared to modern values (Siegert and Payne, 2004; Neumann et al., 2008; Koutnik et al., 2016).

Many numerical ice-sheet models that aim to predict Antarctica's long-term (past and future) contribution to sea-level rise use past ice-sheet reconstructions from after the LGM to guide and evaluate their models (Chavaillaz et al., 2013; DeConto and Pollard, 2016; Bracegirdle et al., 2019). However, even well-used ice-sheet reconstructions assume that the ice sheet retreated continuously throughout the Holocene (e.g. RAISED Consortium, 2014), a finding that has been challenged recently for the WAIS (e.g. Kingslake et al., 2018). Further, significant discrepancies between model simulations and the palaeo-proxy record currently impede our ability to predict confidently how the ice sheet will respond to future changes in the climate (e.g. Johnson et al., 2021). While improvements in model parameterisations are needed to close this gap (Bracegirdle et al., 2019; Sutter et al., 2021), considerable improvement in the availability and quality of palaeo-proxy records, particularly during the Holocene, is also needed to provide better constraints for ice-sheet models and ultimately better resolve past ice-sheet changes (Kingslake et al., 2018; Jones et al., 2022). Palaeo-proxy data have traditionally come from point-based measurements, such as ice cores (e.g. Petit et al., 1999; Parrenin et al., 2007; WAIS Divide Project Members, 2013; Buizert et al., 2021), sediment cores (e.g. Hillenbrand et al., 2013; Arnd et al., 2017; Hillenbrand et al., 2017; Kingslake et al, 2018; Venturelli et al., 2020; Neuhaus et al., 2021; Sproson et al., 2022), or surface-exposure dating (e.g. Stone et al., 2003; Suganuma et al., 2014; Johnson et al., 2014; Hein et al., 2016; Nichols et al., 2019; Johnson et al., 2020; Braddock et al., 2022). A complimentary and spatially extensive alternative data source for inferring past climate across an ice sheet is provided by Internal Reflecting Horizons (IRHs) detected by RES. They primarily result from englacial acidity contrasts and are often detected for hundreds of kilometres on RES data (Harrison, 1973; Bingham and Siegert, 2007). When employed in combination with ice-core stratigraphies, IRHs can be used to extend age-depth relationships away from an ice core by following peaks in electromagnetic return power in the radar data (e.g. Beem et al., 2021; Bodart et al., 2021a; Cavitte et al., 2016; Jacobel and Welch, 2005; MacGregor et al., 2015; Whillans, 1976; Winter et al., 2019).

In contrast to East Antarctica and Greenland, IRH extension of WAIS ice cores has so far been challenging due to fewer deep ice cores there and, until recently, the lack of well-dated IRH datasets. However, efforts have intensified in recent years to improve our understanding of ice stratigraphy over this sector. In particular, four recent studies using airborne RES data (Karlsson et al., 2014; Muldoon et al., 2018; Ashmore et al., 2020a; Bodart et al., 2021a) all identified a distinct and bright IRH dated using the Byrd and WD14 ice-core chronologies to 4.72 ± 0.28 ka BP (Muldoon et al., 2018; Bodart et al., 2021a). A comparison of volcanic sulphate deposition within the WD14 and Siple Dome ice cores revealed a large peak in sulphate concentration matching the age and depth of this ubiquitous IRH (Kurbatov et al., 2006; Bodart et al., 2021a; Cole-Dai et al., 2021; Sigl et al., 2022), which we hereafter term the "4.72 ka IRH". This IRH has now been observed by multiple RES systems and extends throughout much of the slower-flowing ice of the Amundsen and Weddell Sea embayments (< 400 m a$^{-1}$), including across the divides demarcating regions draining into the Amundsen, Weddell and Ross Seas.

Despite their potential wide-ranging applications, the incorporation of IRHs into ice-sheet models has so far been limited compared to other types of palaeo-proxy data, primarily because the inference of accumulation-rate or ice-flow history from IRHs is an ill-posed inverse problem (Waddington et al., 2007). Previous applications using IRHs to inform regional and continental models include: (a) constraining decadal-scale Surface Mass Balance (SMB) estimates from atmospheric models using annually-resolved IRHs found in the shallow firn (Medley et al., 2013; 2014; Van Wessem et al. 2018; Dattler et al., 2019; Kaush et al., 2020; Cavitte et al., 2022); (b) inferring past accumulation rates going back further in time (i.e. 100s to 1000s years) with the aim of comparing past accumulation estimates with modern times (e.g. Leysinger Vieli et al., 2004; Siegert and Payne, 2004; Neumann et al., 2008; MacGregor et al., 2009; 2016; Leysinger Vieli et al., 2011; Cavitte et al., 2018); or (c) integrating both their characteristics (e.g. elevation in the ice) and the information inferred from them (e.g. accumulation or basal-melt rates) to evaluate the output from regional and continental ice-sheet models (Leysinger Vieli et al., 2011; 2018; Holschuh et al., 2017; Sutter et al., 2021). Promisingly, Sutter et al. (2021) recently showed that spatially extensive Antarctic IRHs can provide unique benchmarks for constraining ice-sheet model parameterisations (i.e. climate forcing and simulated ice flow), which are then used to simulate palaeo ice-sheet evolution. Together, these studies indicate multiple avenues for ice-sheet models to assimilate IRHs to further improve estimates of past, current and future ice-sheet changes.

Here, we estimate mid-Holocene accumulation rates across the WAIS from first-order calculations using a one-dimensional (1-D) model, constrained by the spatially extensive 4.72 ka IRH. We first describe the data, the model used and their limitations and uncertainties (Sect. 2). We then present our accumulation-rate estimates and compare them to observed and modelled modern accumulation rates to reveal a longer-term perspective on changes between the mid-Holocene and the present (Sect. 3). Finally, we place our results in the context of previous studies that consider WAIS evolution during the Holocene (Sect. 4).

## 2. Data and methods

### 2.1 Along-track IRH data

We used data from extensive (~91 000 flight-track km) RES surveys acquired across West Antarctica between 2004 and 2018. The main contributing surveys are the University of Texas Institute for Geophysics (UTIG) 2004-2005 AGASEA survey flown over Thwaites Glacier (THW) and Marie Byrd Land which deployed the 60-MHz High Capability Airborne Radar Sounder (HiCARS) radar system (Holt et al., 2006; Peters et al., 2007), and the British Antarctic Survey (BAS) 2004-05 BBAS survey over Pine Island Glacier (PIG) and 2010-2011 IMAFI survey over Institute and Möller Ice Streams (IMIS) which deployed the 150-MHz Polarimetric Airborne Survey INstument (PASIN) radar system (Vaughan et al., 2006; Corr et al., 2007; Ross et al., 2012; Frémand,

Bodart et al., 2022) (Fig. 1; Table 1). Additional profiles from NASA's Operation Ice Bridge (OIB;
MacGregor et al., 2021) 2016 and 2018 surveys, flown with the 195-MHz Multichannel Coherent
Radar Depth Sounder 2 (MCoRDS-2) radar system (CReSIS, 2018), were also used to extract IRH
information near the WD14 Ice Core and upper IMIS catchments (Bodart et al., 2021a; Figure 1 and
Table 1). We refer the reader to the above references for comprehensive details on each system's
capabilities.

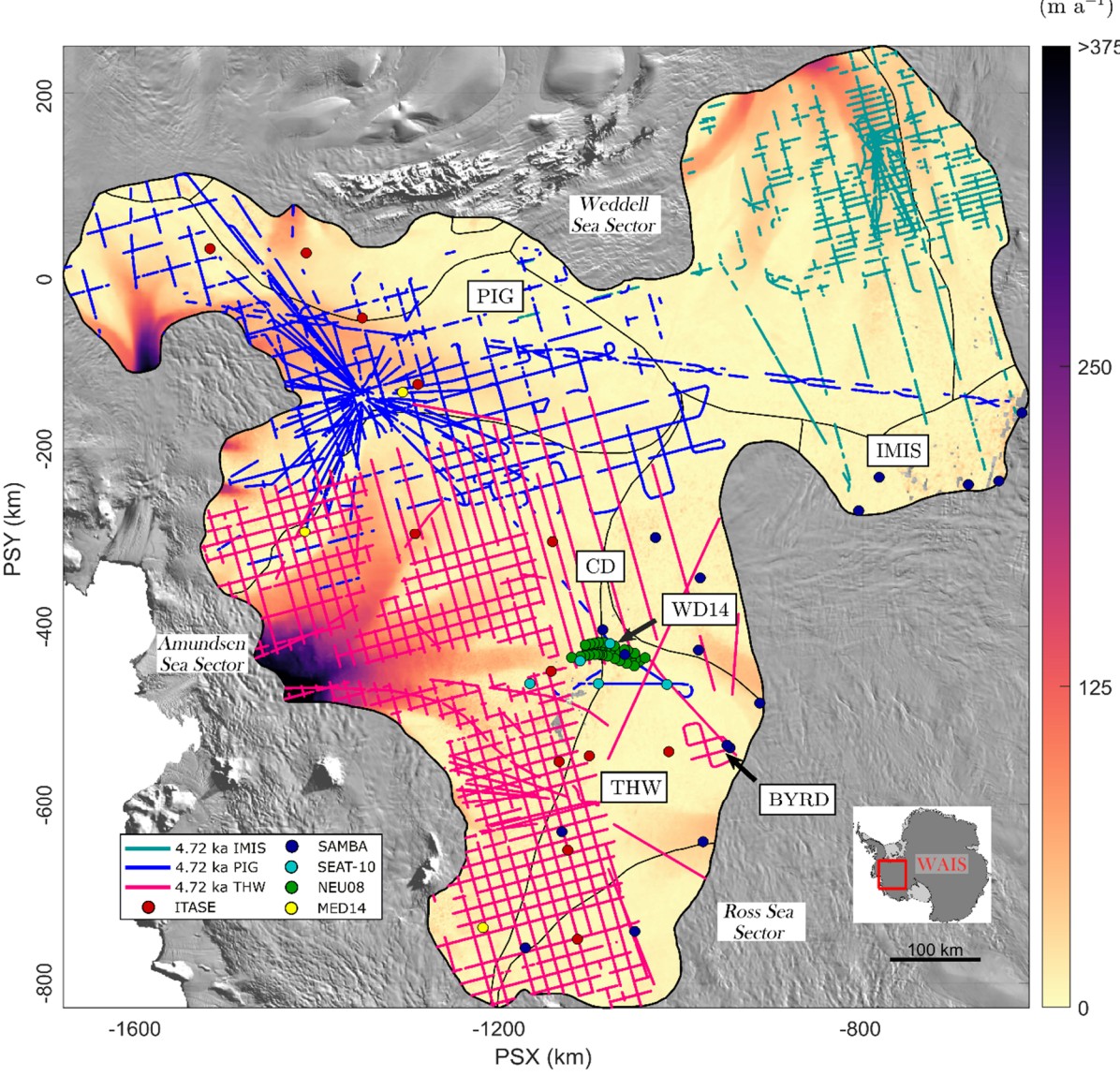

Figure 1. Map of the datasets and key locations in this study. The three datasets that contain the 4.72 ka
IRH are colour-coded as IMIS (green), PIG (blue), and THW (pink). IRH data where $D > 1$ are excluded (see
Section 2.2.1; Figure S1). Points represent the snow, firn and ice cores used in this study to compare modern
accumulation rates with those inferred from the 4.72 ka IRH (Sect. 2.4). The background colour map shows
modern surface speeds from Rignot et al. (2017). Locations mentioned in this paper are abbreviated on the map,
as follows: BYRD (Byrd Ice Core), IMIS (Institute and Möller Ice Streams), PIG (Pine Island Glacier), THW
(Thwaites Glacier), WAIS (West Antarctic Ice Sheet), CD (Central Amundsen-Weddell-Ross Divide), WD14
(WAIS Divide Ice Core). Major ice divides are from Mouginot et al. (2017). The background image is the 2014
MODIS mosaic of Antarctica (Haran et al., 2018). For all analysis and figures in this study, the SCAR Antarctic
Polar Stereographic projection is used (PSX/PSY; EPSG: 3031).
These RES surveys were used to track and date six IRHs spanning the Late Pleistocene and
Holocene (25.7 – 2.3 ka BP) that collectively cover much of the WAIS, including IMIS (Ashmore et

al., 2020a), PIG (Karlsson et al., 2014; Bodart et al., 2021a) and THW (Muldoon et al., 2018). Here we only consider the 4.72 ka IRH mapped in all four studies and shown in Figure 1, as it is by far both the most spatially extensive and the only commonly traced IRH across all studies. We first merged all data points from the 4.72 ka IRH across the three catchments, resulting in a cumulative distance of ~40 000 line-km of IRH profiles (44% of the RES surveys' total coverage; Table 1). Although the along-track RES data were acquired with a trace spacing of between 10 and 35 m, depending on the dataset used, we re-sampled these points to 500 m in the along-track direction. We then added a spatially invariant firn correction of 10 m onto the Muldoon et al. (2018) dataset to match the same firn correction applied by the other studies to correct the IRH depth. Finally, we calculated the median value of all ice thicknesses and IRH depths falling within each 500 m interval.

Table 1. Characteristics of each IRH dataset used in this study that contain the 4.72 ka IRH. 'Reflector 1' in Muldoon et al. (2018) is abbreviated here as 'R1'.

| Survey name | Survey provider | RES system | Dataset reference | Cumulative IRH distance ($10^3$ km) |
|---|---|---|---|---|
| IMAFI | BAS | PASIN 150-MHz | H2 in Ashmore et al. (2020a) | 15 |
| BBAS / OIB | BAS / NASA | PASIN 150-MHz / MCoRDS-2 195-MHz | R2 in Bodart et al. (2021a) | 6 |
| AGASEA | UTIG | HiCARS 60-MHz | R1 in Muldoon et al. (2018) | 19 |

## 2.2 Inferring accumulation rates

To infer accumulation rates from the 4.72 ka IRH, we used the Nye model, a 1-D ice-flow model widely used for estimating accumulation rates and age-depth relationships over relatively slow-flowing parts of an ice sheet (Nye, 1957; Fahnestock et al., 2001a). This model invokes the local-layer approximation (LLA), i.e. it assumes that the time-averaged accumulation rate that the IRH has experienced since its upstream inception at the surface can be adequately represented by its depth where it is observed presently. Other 1-D models exist, including the Dansgaard-Johnsen (Dansgaard and Johnsen, 1969) and the shallow-strain rate model (MacGregor et al., 2016), but were less suitable for estimating accumulation rates here due to uncertainty in the basal shear layer thickness across our survey area and because we are limited to only one IRH to constrain the ice-flow model, respectively. The Nye model assumes that ice thickness is constant and therefore that the ice sheet has been in a steady state since the deposition of the IRH, an acceptable assumption for the period under investigation here. The Nye model states:

$$\dot{b}_a = \ln\left(\frac{z_a}{H}\frac{H}{a}\right), \tag{1}$$

where $\dot{b}_a$ is the mean accumulation rate during the Holocene epoch between an IRH of age $a$ and the present, $z_a$ represents the depth of the IRH dated at the ice core, and $H$ is the ice thickness. The model assumes that the vertical strain rate, $\dot{\varepsilon}_{zz}^a$, is also constant and vertically uniform, so that it exactly balances the overburden of local ice accumulation:

$$\dot{\varepsilon}_{zz}^a = \frac{\dot{b}_a}{H}. \tag{2}$$

We iterated Eq. (1) over the re-sampled 500-m spaced dataset using the depth of the 4.72 ka IRH for $z_a$ and used the median radar-derived ice-thickness measurement re-sampled over the 500-m grid to obtain $H$, when this information was available. In areas where the radar did not sound the bed,

we used the BedMachine Antarctica v2 gridded product to obtain a value for $H$ (Morlighem, 2020;
Morlighem et al., 2020). Note that accumulation rate values presented in this study are all reported in
m a$^{-1}$ of ice equivalent using a density value in ice of 917 kg m$^{-3}$.

### 2.2.1 Assessing the suitability of the 1-D model

To quantify the suitability of the LLA which is used here to estimate accumulation rates, we
calculated the effects of horizontal gradients in modern ice thickness and accumulation rates along
particle paths in their ability to affect IRH depths across our grid, as per Waddington et al. (2007).
Where these gradients are large, estimates of accumulation rates from IRHs likely require a more
complete treatment of ice flow and its effect upon IRH depths, which multi-dimensional models and
more physically complete models can better resolve (e.g. Waddington et al., 2007; Leysinger Vieli et
al., 2011; Karlsson et al., 2014; Nielsen et al., 2015; Koutnik et al., 2016;). However, such models are
significantly more computationally expensive over such a larger area and depend on well-constrained
boundary conditions from along-flow radar profiles which are not often available at an ice-sheet level
(MacGregor et al., 2009).
We quantified the effect of horizontal gradients on an IRH of age $a$ by first estimating the
total horizontal particle path length $L_{path}$ each "particle" of the 4.72 ka IRH has travelled since $a$, and
then the characteristic lengths of variability in ice thickness ($L_H$) and apparent accumulation rate ($L_{\dot{b}}$)
(Supplementary Information). These three components were then combined to generate a non-
dimensional parameter $D$ (Fig. S1d), which we used as a confidence metric for our inferred
accumulation rates. Both Waddington et al. (2007) and MacGregor et al. (2009) suggested a value of
$D \ll 1$ over Antarctica, whereas MacGregor et al. (2016) used a maximum value of $D = 1$ to estimate
where the LLA is acceptable over Greenland. Because $D$ cannot be translated simply into an
uncertainty in an LLA-inferred accumulation rate, it is not yet clear what exact value is appropriate.
Smaller values of $D$ indicate that local horizontal gradients in ice thickness and accumulation rates
have a smaller effect on IRH depth of age $a$, and thus that the LLA may be valid (Waddington et al.,
2007; MacGregor et al., 2009; 2016). Where $D \geq 1$, the depth of an IRH is less likely to be the result
of accumulation rates at the surface or vertical strain rates further down, and thus a more sophisticated
model is likely required (Sect. 2.2.2) (Waddington et al., 2007). However, MacGregor et al. (2009)
found that even along a flowband across Lake Vostok where the mean value of $D$ is 0.50 for a 41-ka
IRH, the difference in accumulation rate inferred from the LLA and from a more sophisticated
flowband model could be relatively small (4-16%). This similarly suggests that accumulation rate can
be inferred acceptably using the LLA where $D$ is higher.
For our study area, $D$ values are mostly well below unity (median: 0.19; 25$^{th}$ quartile: 0.09;
75$^{th}$ quartile: 0.34), which suggests relatively little effect from ice-dynamical processes upon IRH
depths across most of our grid. We used the upper quartile of the $D$ distribution across our model
domain (i.e. $D \leq 0.34$) to show areas where we can have confidence that accumulation rate remains
the dominant factor influencing the vertical position of our IRHs in the ice column (i.e. where the
$D \ll 1$ criterion is likely met; Fig. S1d). While accumulation rates inferred from IRHs situated in the
upper quartile (Fig. S1d) may still be valid, we suggest additional caution in interpreting our results
there due to the potential impact of larger flow gradients on IRH depths.

### 2.2.2 Model limitations and uncertainty

One of the main limitations of the Nye model is that it assumes that gradients in sliding
velocity are mostly concentrated in a thin layer at the ice-bed interface and that the ice column
deforms by pure shear only (Nye, 1957; Fahnestock et al., 2001a). For this reason, the Nye model is
generally only appropriate for IRHs found in the upper part of the ice column, as is the case here.
Additionally, the use of the model is restricted to areas where ice flow is relatively slow and
horizontal strain rates are also relatively low.
Here we focus on a shallower IRH situated in the upper part of the ice column (median: 40%;
$25^{th}$ quartile: 30%; $75^{th}$ quartile: 50%; Fig. 2b-c), for which we can reasonably assume that the ice
sheet has remained close to steady state and where IRHs are likely shallow enough not to have
experienced appreciable flow disturbances that would affect the Nye model assumptions.
Additionally, due to the inherent nature of tracking IRHs through RES data, our coverage is limited to
areas where ice-flow speeds are relatively low and IRHs are undisturbed. In some portions of our
study area, the IRH is found deeper in the ice column or in faster-flowing sections of the ice sheet
(e.g. in the downstream sectors of our grid, Figs. 1 and 2b-c); areas where the assumptions that the 1-
D model is based on may be challenged.
Estimating uncertainty in accumulation rates from the Nye model is non-trivial. Previous
studies have used the misfit between the accumulation rate calculated using multiple proximal IRHs
in the ice column (e.g. Fahnestock et al., 2001a; 2001b; Leysinger Vieli et al., 2004; MacGregor et al.,
2016). Unfortunately, this method is not suitable here due to the dearth of spatially extensive IRHs
younger than 4.72 ka over our model domain.
Instead, uncertainty in the Nye-inferred accumulation rates were calculated using: (a) the
lowest and highest possible accumulation rates from Eq. (1) using the age uncertainty (± 0.28 ka) of
the 4.72 ka IRH and (b) the lowest and highest possible accumulation rates inferred from an additional
1-D model (Eq. S5) which accounts for the effect of strain rates on accumulation rates (i.e. the
shallow-strain rate model from MacGregor et al. (2016); Supplementary Information; Fig. S2-4).
This calculation provides lower and upper bounds for the IRH-inferred accumulation rates
(Fig. S4a-b), which were then averaged to generate a relative uncertainty (Fig. S4c). From this
assessment, we estimate a median relative uncertainty in the Nye-inferred accumulation rates for the
4.72 ka IRH of 14% across our grid. This uncertainty is higher in the downstream edges of our grids,
particularly over the PIG, THW and IMIS catchments, and generally low over the Amundsen-
Weddell-Ross divide (Fig. S4), reflecting the effect of spatially variable strain rates on the inferred
accumulation rates. When combined with the assessment of the suitability of the LLA and exclusion
of IRHs where the $D > 1$ (Sect. 2.2.1-2.2.2), we conclude that it supports our application of a 1-D
modelling approach here.

### 2.3 Gridding and filtering

Once IRH depths and accumulation rates for the 4.72 ka IRH were obtained at regular 500-m
points along RES flight paths, we filtered the results using a moving-average Gaussian filter of length
samples (equivalent to ~15 km) to reduce along-track noise in the IRH depth. We then gridded the
filtered result using a Delaunay-triangulation-based natural neighbour interpolation method onto a 1-
km polar stereographic grid. We further smoothed the gridded data using an 18-km square cell mean
filter to limit the localised interpolation artefacts in areas of poor survey coverage. Figure S5 shows
the maximum distance away from the nearest 500-m along-track point used to produce Figures 2-3,
and thus where errors in the interpolated grids are expected to be larger. The median value of this
maximum distance is 5 km and its maximum value is 75 km, which is comparable to previous studies
that infer SMB from IRHs in the shallow firn (e.g. Medley et al., 2014). We evaluated other possible
interpolation methods (e.g. kriging and using different semi-variogram models), but they resulted in
similar or poorer quality and were thus discounted.

### 2.4 Comparison with modern observations

To compare our inferred accumulation estimates for the past 4.72 ka with modern values
(defined here as 1651-2019), we derived information on modern accumulation rates from two sources,
one modelled (gridded) and one from a series of observational (point-based) datasets.

We used modelled gridded accumulation rates from the Regional Atmospheric Climate Model 2.3p2 (hereafter RACMO2) 1979-2019 SMB product forced at its margin with the ERA-Interim product (native resolution: 27 km) as an estimate for modern accumulation rates (Van Wessem et al., 2018). Although SMB is not technically equivalent to the accumulation rate, runoff and sublimation are negligible in our survey area (Medley et al., 2013) so we assume SMB is equal to accumulation rate in this region. We converted modelled values from $kg\ m^{-2}\ a^{-1}$ to $m\ a^{-1}$ of ice equivalent using an ice density value of $917\ kg\ m^{-3}$, calculated the 40-year mean, and then bi-linearly interpolated the gridded RACMO2 product to the same 1-km grid resolution as our 4.72 ka-to-present accumulation grid (Sect. 2.3) to ensure consistency when comparing both datasets.

Observational point-based measurements were obtained from a series of snow, firn and ice cores from the ITASE (Mayewski and Dixon, 2013), MED14 (Medley et al., 2014), SAMBA (Favier et al., 2013), and SEAT-10 (Burgener et al., 2013) datasets, as well as from a network of centennially-averaged modern accumulation rates derived from shallow IRHs traced on ground-based RES data over the central divide and dated using a shallow ITASE Ice Core (Neumann et al., 2008) (Fig. 1). This resulted in 79 point-based accumulation measurements from cores covering the period 1651-2010 CE (Common Era) and spread across our model domain (see Figure 1). Further detail on these datasets can be found in the above references.

To compare the Holocene gridded product with the point-based measurements, we first calculated the average value of the accumulation rate at the point measurement for the entire period. We converted these values to ice-equivalent accumulation rates and then extracted two paired values, i.e. the value for the point-measurement for modern accumulation rates and the value for the nearest grid cell in the gridded 4.72 ka-to-present accumulation estimates to this measurement.

**Results**

The final grids for depth and accumulation rates for the 4.72 ka IRH are shown in Figures 2 and 3. In total, these grids are made of ~89 000, 500-m spaced points, which cover an area of ~610 000 $km^2$, or ~30% of the total surface area of the WAIS. The grids span most of the PIG and THW glacier catchments, as well as the Ronne (upper Rutford, Institute, and Möller) and upper western Ross (Bindschadler, Kamb, MacAyeal, and Whillans) catchments (IPY Antarctic boundaries G-H, J-Jpp, and Ep-F; Mouginot et al. (2017); Fig. 1-2). Overall, the 4.72 ka IRH is shallower within the IMIS and upper PIG and THW catchments, as well as on the Ross side of the central divide where ice thickness is particularly deep (Fig. 2b). Conversely, the 4.72 ka IRH is deeper in the ice near a 400-m high bedrock plateau that separates the northern and southern basins of PIG (Vaughan et al., 2006) and at two locations in the upstream parts of the main trunk of THW where ice flows over highs in subglacial topography (Fig. 2b).

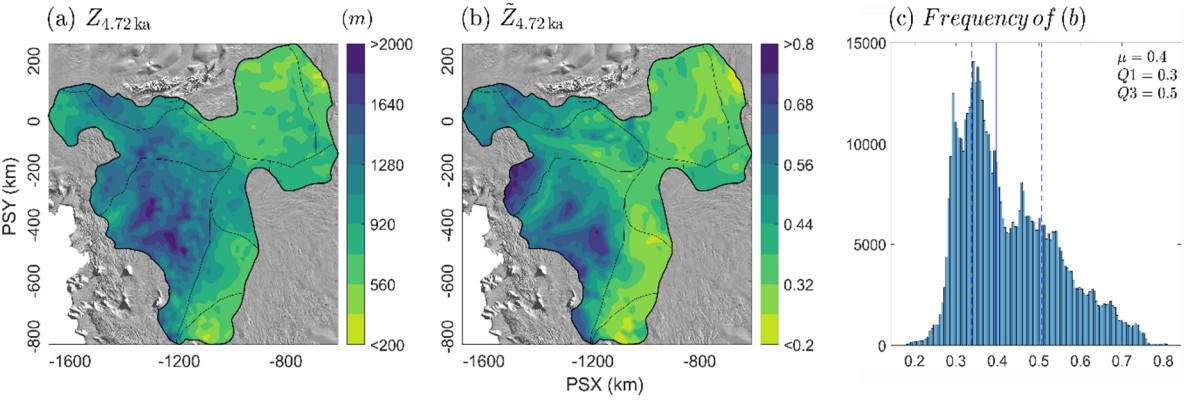

Figure 2. Gridded depths for the 4.72 ka IRH across the model domain covering the PIG, THW, and Institute and Möller ice-stream catchments. (a) Gridded depth of the 4.72 ka IRH. (b) Normalised depth of the

4.72 ka IRH relative to ice thickness. (c) Histogram showing the distribution of values in (b) with the median ($\tilde{\mu}$)
and interquartile range (i.e. 25$^{th}$ (Q1) and 75$^{th}$ (Q3) quartiles) shown as solid and dashed blue lines respectively.
The background image is the 2014 MODIS mosaic of Antarctica (Haran et al., 2018).

### 3.1. Catchment-scale accumulation estimates

Figure 3 shows a comparison of the ice-equivalent accumulation rates we inferred for the 4.72
ka IRH (Fig. 3a) and modern SMB estimates from RACMO2 (Fig. 3b). We observe that the IRH
accumulation rate pattern for the last 4.72 ka is similar to the modern pattern of accumulation rates for
the Amundsen Sea sector of the WAIS, which is dominated by higher coastal accumulation rates that
progressively decrease inland to reach their lowest rates over the Ross side of the divide (Fig. 3a-b).
Differences in accumulation rates between the 4.72 ka-to-present estimates and modern values are
mainly observed directly upstream of the main trunks of PIG and THW, where modern rates are much
higher (up to 0.2 m a$^{-1}$ ice equivalent) than for the 4.72 ka-to-present estimates (Fig. 3c). In
comparison, higher accumulation rates for the last 4.72 ka relative to modern rates are observed for
the entire stretch of the Amundsen-Weddell-Ross divide (Fig. 3c; Table 2). Over the IMIS catchment,
little change is observed between the two periods. Over the entire model domain, we observe a
median percentage change value of 6% higher accumulation since 4.72 ka compared with modern
rates (Fig. 4); however, when considering only the values that fall within 100 km of either side of the
Amundsen-Weddell-Ross divide (i.e. in the accumulation zone of the Amundsen, Weddell, and Ross
Sea sectors and where mean surface speeds average ~7 m a$^{-1}$), we obtain a median percentage change
value of 18% higher accumulation compared with modern accumulation rates (Fig. 4).

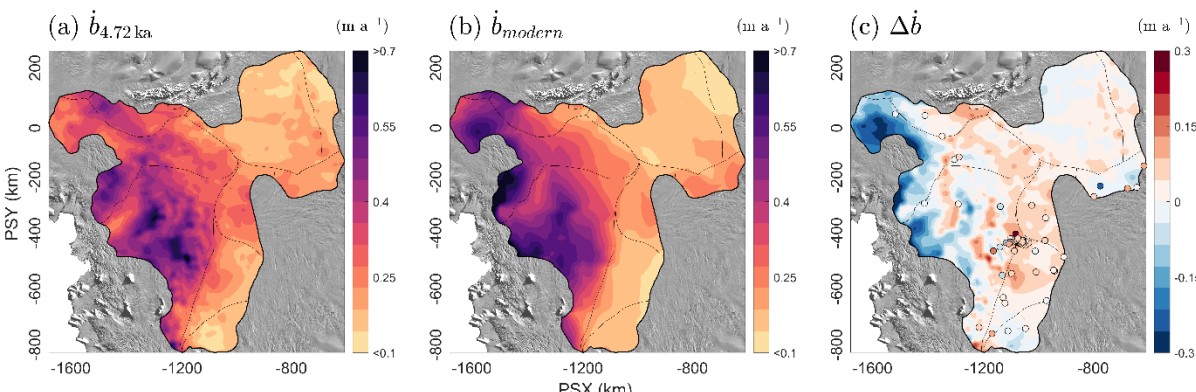


Figure 3. Gridded estimates of ice-equivalent accumulation rates for the last 4.72 ka and modern times.
(a) Gridded accumulation rates inferred from the 4.72 ka IRH. (b) Modern (1979 – 2019) modelled SMB rates
from RACMO2. (c) Difference between 4.72 ka-to-present and modern accumulation rates (red = 4.72 ka-to-
present accumulation higher than modern times, blue = 4.72 ka-to-present accumulation lower than modern
times). The dots represent the difference between the value for the nearest grid cell in (a) and time-averaged
accumulation rates at each of the 79 core locations (see Section 2.4; Fig. S6). The background image is the 2014
MODIS mosaic of Antarctica (Haran et al., 2018).
Comparison between our 4.72 ka-to-present accumulation-rate estimates and 79 core-derived
point-based accumulation measurements for modern times (1651-2010 CE) are shown in Figures 3-4
and S6. This evaluation shows that the 4.72 ka-to-present accumulation-rate estimates for the nearest
grid cell to each point measurement are, on average, 22% higher for cores situated across the entire
grid ($p$ < .0015, $n$=79) and 23% higher for cores found within 100 km of the divide compared with
modern accumulation rates ($p$ < .0001, $n$=59; Figs. 4 and S6). In comparison, a similar analysis
between grid cells from the 4.72 ka-to-present accumulation-rate estimates and RACMO2 at these 79
core locations shows mid-Holocene accumulation rate estimates are, on average, 32% ($P$ < .00002,
n=79) higher for cores situated across the entire grid and 36% higher for cores found within 100 km
of the divide ($p$ < .00001, n=59; Fig. S6). This result confirms that the relative change for gridded
accumulation rates between the 4.72 ka-to-present and modern modelled accumulation rates is
consistent with modern rates from point-based measurements.
Table 2. Summary statistics for the modern (modelled and observational) and 4.72 ka-to-present ice-
equivalent accumulation rates at the catchment-scale and over the Amundsen-Weddell-Ross divide (abbreviated
CD for Central Divide here). Values for the Amundsen-Weddell-Ross divide are for all points that fall within
100 km of either side of the divide (see dashed line in Figure 4). $\tilde{\mu}$ refers to the median and IQR represents the
Interquartile Range calculated by computing the difference between the 75$^{th}$ and 25$^{th}$ percentiles. Note that the
values provided in the text represent the median relative change from the cell-by-cell change between each grid
(Fig. 4), rather than the relative change of the median values provided here.

| Accumulation rate (m a$^{-1}$) | *Catchment-wide* | | *CD only* | |
|---|---|---|---|---|
| | $\tilde{\mu}$ | *IQR* | $\tilde{\mu}$ | *IQR* |
| *Modern (model)* | 0.23 | 0.23 | 0.22 | 0.10 |
| *Modern (cores)* | 0.24 | 0.12 | 0.24 | 0.09 |
| *4.72 ka-to-present* | 0.27 | 0.18 | 0.27 | 0.11 |


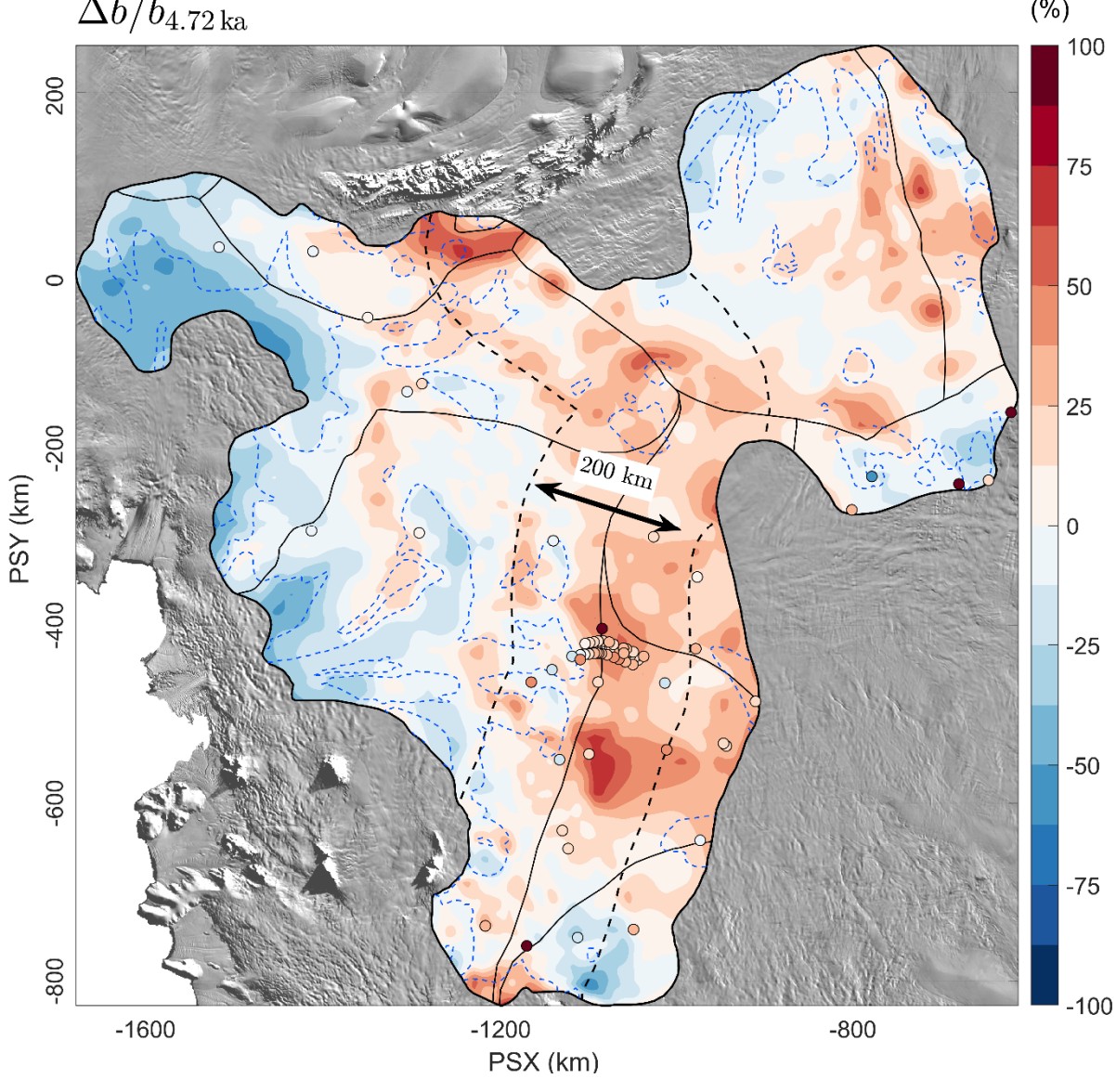


Figure 4. Relative change in accumulation rates between the 4.72 ka-to-present estimates and modern rates. The points on the map represent the relative change in ice-equivalent accumulation rate between the nearest grid cell in the 4.72 ka-to-present grid and the 79 modern observations from cores (Figs. 1 and S6; Sect. 2.4). The dashed black outline line represents the 100-km boundary on either side of the Amundsen-Weddell-Ross divide used to provide the summary statistics in Section 3.1 and Table 2. The dashed blue line shows the contours of the upper limit of the interquartile range for the $D$ parameter ($D \leq 0.34$), whereby all values situated inside of this boundary may satisfy the $D \ll 1$ criteria and those outside may require re-evaluating with the use of multi-dimensional models (Sect. 2.2.1-2.2.2). The background image is the 2014 MODIS mosaic of Antarctica (Haran et al., 2018).

## 3.2 Elevation-dependent accumulation estimates

While Figures 3 and 4 help to assess potential differences in patterns and rates across spatial scales, considering accumulation-rate differences in terms of elevation can inform how topography influences accumulation and whether this has changed over time. We binned the ice-equivalent accumulation values by 50-m elevation bands across the three main catchments covering our grid (Amundsen, Weddell and Ross) for both the 4.72 ka-to-present estimates and modern model rates and calculated the mean accumulation rate and the total accumulation rate for each bin over the entire elevation gradient (Fig. 5). We again find that the accumulation-rate estimates for the period since 4.72 ka are lower at low elevations (~700 – 1400 m) over the Amundsen sector compared with RACMO2, but begin to exceed RACMO2 near the 1400-m elevation band where the 4.72 ka-to-present accumulation rate is higher than modern times across the divide (Fig. 5a-b). We also note that whilst an elevation-dependent gradient in accumulation rates, dominated by high accumulation at the coast decreasing inland, exists over this sector for the mid-Holocene, it is much less marked than for present rates. This is not surprising, as this sector is where we observe the largest relative uncertainties in inferred accumulation rates across our grid (Fig. S4), indicating that the 1-D model is less able to produce realistic accumulation rates in the downstream end of our grid where ice flow is faster and strain rates are likely higher. In comparison to the Amundsen sector, accumulation rates since 4.72 ka are generally higher at all elevations for the Weddell and Ross sectors compared with the present, although this difference is less than over the Amundsen sector (Fig. 5 c-f).

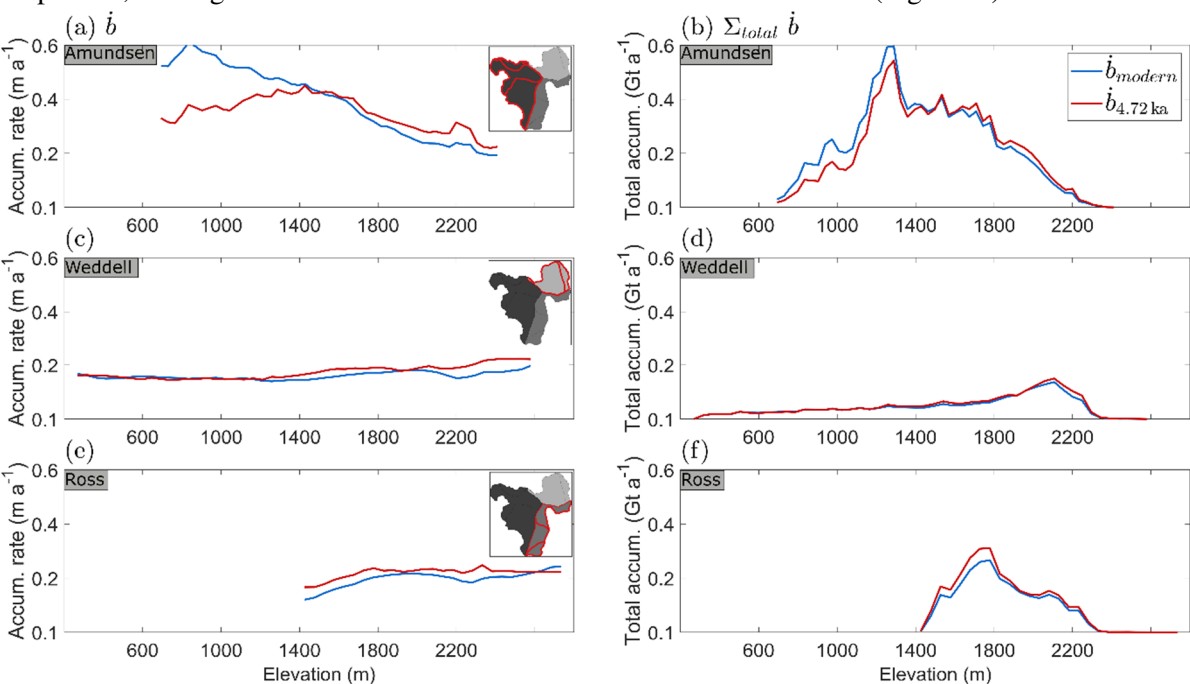

Figure 5. Comparison of ice-equivalent accumulation rates between the 4.72 ka-to-present estimates and modern rates (RACMO2) binned by 50-m elevation bands across the three main catchments considered here (Amundsen, Weddell, and Ross). (a, c, e) Mean accumulation rate averaged per 50-m elevation band across the

specific catchment area in m a$^{-1}$. (b, d, f) Total accumulation rate per 50-m elevation band across the specific
catchment area in Gigatonnes per annum (Gt a$^{-1}$).

## 4. Discussion

### 4.1. Comparison with other Holocene accumulation estimates

Previous studies of past accumulation rates over the WAIS have shown that accumulation varied
temporally during the Holocene. Using a single airborne RES profile over the Amundsen Sea sector,
Siegert and Payne (2004) showed that accumulation rates were approximately the same at 3.1 ka
compared with modern rates, but ~0.3 m a$^{-1}$ greater (~15 %) than current rates between 3.1-6.4 ka,
before which accumulation was ~50% of modern rates between 6.4 and 16.0 ka. Similarly, Neumann
et al. (2008) found that accumulation rates at the Amundsen-Weddell-Ross divide were ~30% higher
between 3-5 ka than modern values based on a dense network of IRHs traced on ground-based RES
data, while Karlsson et al. (2014) found that accumulation patterns had likely changed twice during
the early to mid-Holocene over PIG from the lack of a model fit between the depths and ages of two
prominent IRHs. Using the updated WD14 record, Fudge et al. (2016) showed that accumulation rates
were higher there in the mid to late-Holocene (19% between 4.72 ka BP and the present), a trend that
was also observed by Koutnik et al. (2016), who found a 20% increase in accumulation rates between
2-4 ka compared with modern rates from a ground-based RES profile across the ice divide.
These studies together point to a period of increasing accumulation observed at the WD14 Ice
Core from ~7 ka onwards (Fudge et al., 2016; their Figure 2), with its peak matching the age of the
4.72 ka IRH used here. Thus, our accumulation-rate estimates likely form part of a wider pattern of a
sustained increase in accumulation across the Amundsen-Weddell-Ross divide over several millennia.
In showing that mean accumulation rates since 4.72 ka were 18% greater than modern rates modelled
from RACMO2 across the Amundsen-Weddell-Ross divide, our results provide much wider regional
support for the hypothesis that accumulation rates during the mid-Holocene exceeded modern rates
across central West Antarctica. A possible explanation for the higher accumulation rates during the
mid-Holocene compared with modern values is that they represent a continued climatic transition
from the LGM (Steig et al., 2001). Alternatively, it has been suggested that seasonal or interannual
variability, such as a weaker circumpolar vortex (van Den Broeke and van Lipzig, 2004; Neumann et
al., 2008), or teleconnections to tropical Pacific Ocean warming (Sproson et al., 2022), may also lead
to such difference. We did not find evidence for significant changes in accumulation patterns between
the mid-Holocene and modern times, suggesting that the current spatial pattern of high accumulation
on the Amundsen side of the divide transitioning to low accumulation on the Ross side of the divide
was stable throughout the mid-Holocene, as previously suggested by others (Siegert and Payne, 2004;
Neumann et al., 2008; Koutnik et al., 2016).
We also find that accumulation estimates for the 4.72 ka-to-present are smaller than modern rates
in the lowest elevation bands (<1400 m), particularly over the Amundsen Sector (Fig. 5 a-d). This
pattern was also found by Medley et al. (2014), who compared modern observational and modelled
data over this sector and hypothesised that this discrepancy at low elevations resulted primarily from a
lack of sufficient accumulation measurements in the lower sections of their survey area. In our case,
these low-elevation values are close to the boundary where we consider the LLA acceptable for the
4.72 ka IRH, albeit where $D$ values are higher than for the rest of the catchment (Figure S1d), so it is
more likely that accumulation rates calculated there are affected by ice-flow gradients and their
influence upon IRH depths leading to lower accumulation rates there. Despite this caveat, Figures 5b
and 5d show that values at low elevations contribute relatively little to the total accumulation (by
mass) over our survey area.
We suggest that future ice-sheet modelling studies investigate the difference in accumulation rates
inferred from our 1-D model using multi-dimensional flowband models to assess effects of divergent
and convergent flow on IRH depth and ultimately accumulation rates, as previously considered
elsewhere in Antarctica (MacGregor et al., 2009). This could be conducted along a flowline
transitioning from the slow-flowing regions directly downstream of the Amundsen-Weddell-Ross
divide to the coastal margins of our grid, particularly over THW where we observe the largest
uncertainties in accumulation rates. In addition, we suggest that future modelling studies use the
accumulation-rate variability from the WD14 Ice Core as a climate forcing in their ice-sheet models.
Koutnik et al. (2016) previously showed that the WD14 record is unique in that it provides a reliable
record of accumulation-rate variability during the Holocene, which other East Antarctic ice-core
records often used to reconstruct the evolution of the WAIS do not possess. We found that these
higher accumulation rates are spatially extensive across nearly one third of the WAIS, further
suggesting that the WD14 Ice Core is indeed representative of the wider WAIS and can be used in
regional or continental ice-sheet models as a reliable climate forcing for the region. Future regional
and continental ice-sheet models should make use of this record to adjust their climatic boundary
conditions to provide improved estimates of ice-elevation change and grounding-line evolution over
Antarctica.
**4.2 Impact for ice-sheet elevation change during the Holocene**
Model results from Steig et al. (2001) suggest that the maximum elevation of the WAIS was most
likely reached during the early to mid-Holocene (around ~7 ka) following higher accumulation rates
at the late glacial–interglacial transition, after which the WAIS slowly declined to present conditions
as the sea-level-rise-induced kinematic wave reached the ice-sheet interior and outpaced the increase
in accumulation rates. However, higher accumulation rates in the mid-Holocene relative to the
present, which our results suggest occurred spatially across the WAIS, would likely delay the timing
of this thinning by several thousand years (Steig et al., 2011).
Using a flowband model, Koutnik et al. (2016) suggested that an increase of up to 40% in
accumulation rates for the period 9 – 2 ka would likely have led to an increase in ice thickness of tens
of metres during the mid-Holocene. Although this finding was warranted by physical assumptions
around the response time of the ice-sheet interior to adjust to an increase in accumulation in the
model, it points to the potential for the divide to have thickened by several metres over a relatively
short period of time from increased accumulation rates alone. Because the WAIS is also sensitive to
ice-dynamical changes at the ice-sheet margins (e.g. grounding-line retreat or calving), an increase in
accumulation rates in the upper part of the ice sheet may not necessarily result in enough thickening to
counteract potential dynamical losses farther downstream (Jones et al., 2022). Conway and
Rasmussen (2008) reported that the Amundsen-Ross Divide is currently thinning and migrating
towards the Ross Sea at a speed of 10 m a$^{-1}$, but they were unable to determine whether this was in
response to long-term (last two millennia) accumulation-rate changes there or short-term (last few
centuries) ice-dynamical forcing from the coastal margins of the Amundsen and Ross sectors. More
recently, Balco et al. (2023) showed that Thwaites and Pope glaciers experienced 35 m of thickening
in the mid-to-late Holocene, when accumulation rates were higher than present. While this thickening
relative to present was attributed to glacio-isostatic rebound, it is also possible that higher
accumulation rates in the upstream sections of the WAIS contributed to this thickening, if sustained
over millennia.
The lack of an ice-dynamical component in the model used here precludes us from evaluating
any ice-surface-elevation change associated with changing accumulation rates. However, 18% higher
accumulation rates during the mid-Holocene relative to the present across 30% of the WAIS could be
consistent with an elevation increase of several tens of metres in ice thickness, according to Koutnik
et al. (2016). Even if tens of metres of ice-surface-elevation change occurred, it is still unlikely to
significantly affect the steady-state assumption of the 1-D model used here (constant ice thickness

over time), because such changes are small (a few per cent of the ice thickness) and that ice thickness exceeds 3500 m in places over our survey area.

### 4.3 Impact for grounding-line evolution during the Holocene over the WAIS

Finally, we consider the possibility for Holocene ice thickening at the divide from increased accumulation rates to affect downstream grounding-line evolution. Recent evidence from ice-sheet modelling and field measurements suggest that grounding-line retreat during the Holocene was not monotonic, particularly at the Ross and Weddell Sea sides of the WAIS (Bradley et al., 2015; Kingslake et al., 2018; Neuhaus et al., 2021). Rather, Kingslake et al. (2018) showed that the grounding-line position in the Ross and Weddell Sea sectors initially retreated from the LGM inland until ~10.2 – 9.7 ka, and then readvanced to its modern position sometime during the Holocene. Although they attributed this change in grounding-line position to the solid Earth viscoelastic response due to ice-sheet mass change and the subsequent re-grounding around pinning points, it has also been suggested that an increase in accumulation rates upstream of the grounding line could lead to a readvance via ice thickening there and a subsequent increase in ice flow (Steig et al., 2001; Koutnik et al., 2016; Jones et al., 2022). Across parts of the Weddell Sea Embayment, several studies have produced evidence for stability of the LGM ice thickness there until the early to mid-Holocene (Ross et al., 2011; Hein et al., 2016; Ashmore et al., 2020a), contrary to most of the WAIS, after which abrupt thinning of ~400 m contributed ~1.4 – 2 m of sea level rise (Hein et al., 2016). A possible explanation for this delayed thinning in the Weddell Sea Embayment is that increased snowfall in the upper WAIS might have counteracted ice-dynamical processes at the coast until the mid-to-late Holocene (Hein et al., 2016; Spector et al., 2019). Similarly, over part of the Ross Sea sector, Neuhaus et al. (2021) showed that the grounding line over Whillans, Kamb, and Bindschadler ice streams retreated to its minimum Holocene position in the mid to late-Holocene, and then readvanced between 2 – 1 ka, coinciding with periods of warmer and colder climates, respectively. They concluded that the reported grounding-line migration was likely dominated by modest climate-induced changes upstream rather than ice dynamics further downstream, as suggested for the Weddell Sea sector (Hein et al., 2016).

Our results, which provide strong and widespread evidence for higher accumulation along the Amundsen-Weddell-Ross divide during the mid-Holocene compared with the present, support these hypotheses further, as higher accumulation rates at the divide would likely result in upstream thickening (Sect. 4.2). In the absence of ice-dynamical processes counter-balancing this increase in accumulation rates, the grounding-line should advance in these regions. However, we note that the pattern of grounding-line retreat and readvance has not been observed over the Amundsen Sea sector (Kingslake et al., 2018; Johnson et al., 2020; 2021; Braddock et al., 2022) despite the accumulation-rate increase we also observed along the Amundsen-Weddell-Ross divide and the recent results from Balco et al. (2023). This complication may indicate that the Amundsen sector is more strongly influenced by coastal changes in ice dynamics, for which even moderate changes in accumulation rate cannot compensate.

## 5. Conclusion

Using a ubiquitous internal reflecting horizon found across most of the Pine Island, Thwaites, and Institute and Möller ice-stream catchments, we have estimated mid-Holocene accumulation rates in the relatively slow-flowing parts of West Antarctica, representing 30% of total surface area of the WAIS.

By comparing our Holocene accumulation-rate estimates with a modern climate reanalysis model and observational syntheses, we estimated that accumulation rates across the Amundsen-Weddell-Ross Sea divide since 4.72 ka were, on average, 18% higher than modern values. Our results suggest that spatial patterns of accumulation across the WAIS have remained stable during this period, i.e.

higher accumulation rates on the Amundsen side of the divide transitioning to lower accumulation rates on the Ross side of the divide. The higher accumulation rates reported here for the mid-Holocene compared to the present agree well with earlier, spatially-focused studies of accumulation rates, all of which indicate higher accumulation rates (+15 - 30%) over the past ~5 ka. This change in magnitude occurred at a time of asynchronous grounding-line migration over the WAIS, including readvances of the grounding line in the Weddell and Ross sectors and evidence for delayed deglaciation in the Weddell Sea side of the WAIS.

The higher mid-Holocene accumulation estimates inferred here over large sectors of the WAIS occurred at a time of sustained, millennial-scale increase in accumulation rates found at the WAIS Divide Ice Core. This pattern indicates that the ice core is suitably representative of the climatic conditions of the wider region over time. We suggest that future regional or continental ice-sheet modelling studies base their palaeoclimate forcing on modern spatial SMB products that are modulated over time using the WAIS Divide Ice Core record. This will enable those models to obtain a more realistic climatic forcing representative of the past conditions of the wider WAIS, and ultimately, constrain ice-sheet volume change and grounding-line evolution during the Holocene.

**Code availability**

All the codes used to produce the results presented in this paper are available on the GitHub page of Julien A. Bodart (https://github.com/julbod, last accessed: 15 March 2023) and on Zenodo (Bodart et al., 2023).

**Data availability**

The IRH information for each of the three surveys used in this paper are archived in open-access repositories (Ashmore et al., 2020b; Bodart et al., 2021b; Muldoon et al., 2023) with references and links provided in the reference list. The BAS airborne radar data which were used to extract the IRHs used in this paper are fully available at the UK Polar Data Centre via the Polar Airborne Geophysics Data Portal (see Frémand, Bodart et al., 2022). The RACMO2 product is available on request from j.m.vanwessem@uu.nl or m.r.vandenbroeke@uu.nl. Links to access the observational point-based datasets used here are available from the respective references mentioned in the text (Section 2.4). The gridded depth and accumulation output from this study are archived on Zenodo (Bodart et al., 2023).

**Author contribution**

J.A.B. designed the study with supervision from R.G.B., D.A.Y., and D.D.B. J.A.B performed the data processing, gridding, and 1-D modelling, with contributions from J.A.M. for the modelling approach. J.A.B. interpreted the results with input from R.G.B., D.A.Y., D.D.B., and J.A.M. J.A.B. wrote the paper, with edits from R.G.B., D.A.Y., J.A.M., D.W.A., E.Q., A.S.H., D.G.V., D.D.B.

**Competing interests**

The authors declare that they have no conflict of interest.

**Acknowledgments**

The authors would like to dedicate this work to our dear friend and colleague, Professor David Vaughan, who recently passed away. This study was motivated by the AntArchitecture SCAR Action Group. UTIG acknowledges the high school students who did the original AGASEA layer interpretation. We would like to thank the editor, Olaf Eisen, as well as Michelle Koutnik and an anonymous reviewers for thorough and constructive reviews, which improved this manuscript.

## Financial support

J.A.B. was supported by the NERC Doctoral Training Partnership grant (NE/L002558/1), hosted in the Edinburgh E$^3$ DTP programme. J.A.B. also acknowledges the Scottish Alliance for Geoscience, Environment and Society (SAGES) for funding a Postdoctoral and Early Career Researcher Exchanges scheme to UTIG. Support for UTIG data analysis was received from NSF grant nos CDI-0941678, PLR-1443690, and PLR-10437661, as well as the G. Unger Vetlesen Foundation and the UTIG Gale White and Ewing/Worzel Fellowships. This is UTIG contribution No. XXX (TBD) and ITGC Contribution No. XXX (TBD).

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
