# Peer review of "High mid-Holocene accumulation rates over West Antarctica inferred from a pervasive ice-penetrating radar reflector"

_The Cryosphere, 2022_

## Referee Comment (RC1)

In this work, the authors perform a widespread tracing of a ~4.72ka layer across significant region of West Antarctica. This is a valuable data set and the authors robustly show how it can be used with simple models to infer the average accumulation conditions around this time. The paper is clearly written and the outcomes clearly presented. I enjoyed reading this contribution and I have mostly minor comments to consider.

One potentially more major consideration, however, is that the lead statement in the abstract, "Modelling the past and future evolution of the West Antarctic Ice Sheet (WAIS) to atmospheric and ocean forcing is challenged by the availability and qulaity of observed paleo-boundary conditions", sets up this work for it's value providing a new constraint on models. I agree that this is an important contribution, but more specific statements about how this new layer product and new inferences of past accumulation can be (or could be) used in models would be helpful. For example, it isn't clear to me if these constraints are valuable to regional ice-sheet models, continent-scale ice-sheet models, and/or also climate models. How could these outputs and inferences practically be used in models, and how many of the current generation of models are set up to use constraints like this in the way they have been shared? For example, few ice models assimilate layers, but is that the emphasis here – that even on Holocene timescales that layers are an important constraint on models, especially in the ice-sheet interior? Or, is it more to advocate using simple models to infer accumulation histories that can then more directly be used as boundary conditions in a wider suite of models? (Or, both!) If there are more specifics that can be added then I think it could have more impact on the modeling community. Some points are mentioned in the manuscript, but structuring parts of the manuscript around this more specifically would be worthwhile if it remains a main motivation and a main conclusion.

Line 25 (related to above point): If possible, I would suggest trying to clarify this point so it is more directly speaking to the modeling community. How do these results advance what is required for model spin up, and if these are continent-scale models then why is this region of West Antarctica and over this time so critical to improve model spin up? Minor rephrasing and a few more words could help make this a more impactful point

Line 38: Since mention comparison to modern in this sentence, could indicate time range over which 18% increase occurs during the mid-Holocene – would help clarify why increase of this amount is important (compared to seasonal or inter-annual variability, for example)

Line 42-44: In general I feel like this point could be a bit more developed, especially if these results are meant to motivate ice-sheet modelers to use new records like this. There may not be space, but a few more words on why modeling past sea-level rise is important could be worthwhile. Elsewhere present and future sea level are also mentioned.

Line 54: Sentence is about modeling changes in ice volume and GL position, so wonder if some more recent references are worth including

Line 56: This is a personal reaction, but the paper has a number of acronyms so it could be worth considering to keep only those that are necessary. I think "GL" could just be given as "grounding line", but again that may just be personal preference and I leave it to the authors to decide! [As an example, it became challenging to read in the text around Figure 1 – as many of these acronyms come into the text – but not sure the best way to handle that.]

Also, could specify what grounding line you are referring to here

Line 71: Could cut "At present"? Perhaps expand this point a bit to clarify if you mean climate models, ice-flow models, or both?

Line 76: By "model physics" do you mean our physical understanding and how that can be represented in models? That could be through parameterization, but guessing that you are also indicating that some physics of the system are still unknown. Could be a bit more clear.

Line 82 and 83: In this list of paleo records, are all of these available to inform WAIS? This paragraph could indicate what motivates the problem generally vs. specifically for West Antarctica. It would help to put in the context of whether the models are regional or continent-scale.

Line 85: Could explain a bit more why radar is an "alternative data source"

Line 92: It isn't completely clear what is meant by "suitable datasets" – entirely an issue with the radar data?

Line 144: Since this is where the data sets are being introduced, could consider stating age range that they cover even more explicitly

Line 184-: Perhaps it is worth mentioning that these layers are well dated, which in this calculation of the relative uncertainty is what leads it to being a low estimate? Or, consider to bring the point up front in the paragraph about this not accounting for uncertainties in the model approximation. 3.3% uncertainty seems low so it could help the reader to explain that more and perhaps define more clearly what the relative uncertainty is

Line 194: I think that MacGregor et al. 2016 only used the LLA (but it would be worth double checking)

Line 200-: "...but because the value of D..." –I think that I understand the point being made, but I think it could be confusing so suggest revisiting

Line 205: This sentence could indicate that a vertical strain correction is still applied based on the model you apply ("sole result of accumulation rates at the surface" doesn't capture that)

Not sure if/where it belongs, but could also considering citing Leysinger Vieli et al. (2011), JGR Earth Surface 10.1029/2010JF001785 for their evaluation of the applicability of vertical flow models to infer accumulation rates from East Antarctic layers (though to find that a 3-D model was necessary)

Could also consider citing Nielsen et al. (2015), Annals of Glaciology 56 (70), 70-78 as while they also find that a 2-D model is more appropriate they also compared a 1-D approach

Line 204: I think that MacGregor et al. 2009 applied a flowband model, not only the LLA so perhaps clarify why this is cited here; can see that Waddington et al. 2007 is cited in reference to the LLA itself (though they also applied a flowband model)

Line 212: "For this reason, the Nye model is generally only appropriate for IRHs found in the upper part of the ice column, as is the case here" – part of the value of using something like the LLA is that it can let you know where you may be surprised by relatively shallow layers (in the upper part of the ice column) but where the Nye model is not appropriate. This point made opens to this, but perhaps could be a chance to reiterate that it may need to be evaluated?

Line 239: Is it necessary to clarify / explain that previous study cited (Medley et al., 2014) was inferring accumulation rates on different timescales and using firn layers – right?

Line 298: Can anything more be shared about why 100 km was picked?

Line 312: Are these numbers the same as ones given in lines 299? I got confused by how these two points were stated

Line 315: Perhaps indicate that point based are "cores", since that terminology comes up later. I had to read this sentence twice so perhaps stating even more explicitly (though it does make sense)

Line 338: Maybe use another phrase than "As a result..."

Has RACMO2 modern or RACMO2 estimates for past decade(s) been evaluated against any ice-core data?

Around Line 386: Is it worth commenting on what this means for recent signal of divide migration (Conway and Rasmussen, 2008)

Line 399: The point "…a finding that must be considered by future modeling studies that simulate past sea-level rise from Antarctica since the LGM" could be strengthened to more directly speak to the modeling community (see above point)

Around Line 415: The interior thickness response to an increase in accumulation in Koutnik et al. (2016) is also tied to assumptions about how fast the limited portion of the ice sheet interior in the model responds to changes in accumulation. This was incorporated in the model based on a physical assumption of ice-sheet adjustment, but it is still an assumption (could see Koutnik and Waddington, 2012). So, while the point made here is reasonable overall, I would suggest emphasizing that tens of meters adjustment depends on the model and also depends on the initial state – the adjustment through mid-Holocene may be sensitive to initial state (early Holocene) and accumulation history for previous few thousand years. And, good that you point out that dynamic component could be important and hasn't been directly taken into account.

So, the point made that "This potential increase in surface elevation is unlikely to affect the steady-state assumption of the 1-D model used here…" isn't necessarily founded on a suite of time-dependent flowband model runs that really tell you whether 10s of meters is the right range. One run gave this range, and bigger elevation change isn't necessarily expected, but this one model run doesn't really address the range of unknowns. Just a comment! Also, may want to clarify that you mean steady-state geometry.

Line 418: I would suggest making a separate paragraph that is directed at the modeling community. This point is a good one, but stating more specifically how this could/should go would be helpful (I think)

Continent-scale modeling from LGM to present does estimate elevation change since the last glacial maximum but there are big discrepancies between models, and this problem isn't reconciled yet (though Jessica Badgeley who recently received her PhD at University of Washington is investigating this). All to say that I think not only mid-Holocene to present elevation change is important, but the potentially bigger changes considered from LGM to Holocene need to be reconciled between models and evaluated with available data (ice-core records, radar data). Is this important to address here – why the emphasis on capturing the mid-Holocene in models compared to the transition from LGM to Holocene (I think ice-sheet adjustment times work as an explanation, but could consider if you need to address that).

Lines 423-433: Is there more of a takeaway related to this work that should be made in relation to these points?

Lines 434-457: This is interesting overall, but wonder if the framing in relation to this new work should be mentioned more up front in the paragraph. It was hard to put into context where this was going until about mid-paragraph. And, the rationale for connecting increased accumulation to interior thickening to GL advance seemed

a bit tenuous given what was shared previously about not really knowing if there was thickening or thinning. The way I read this, there was a bit of back and forth in the discussion points between what may or may not be interpretable.

Would it be more impactful to consider structuring the discussion a bit more around how open questions may be addressed using available constraints, results from this work, and different types of models – what is really the path forward?

Is there a way to strengthen the point in lines 456-457, "potentially indicating that this sector is more controlled by changes in ice dynamics for which even moderate changes in accumulation rate cannot compensate" – in this case why would dynamic thinning not lead to a change in divide position, which may further complicate recovering an elevation change signal? Again, tying these points back into what is known and unknown may help.

Line 472: "making it a powerful dataset for ice-sheet models" – could be more explicit if you mean the layer data or the accumulation-rate inference? I think this is referring to the inference, but is this a data set?

How this becomes a powerful new constraint could be elaborated and clarified what classes of models would be the target. I am also not sure that the closing sentence connects clearly back to what is done here. What models use a "fixed Last Glacial Maximum value" (are these climate, ice, or both) and why does 4.72 ka constraint become so important  from LGM to present – in practice what is being provided to (and suggested of) the modeling community? (see points above)

---

## Author Comment (AC1)

Julien Bodart
School of GeoSciences
University of Edinburgh,
EH8 9XP, UK

Thursday, March 2$^{nd}$, 2023

**Re: [Paper ID #tc-2022-199] High mid-Holocene accumulation rates over West Antarctica inferred from a pervasive ice-penetrating radar reflector by J. A. Bodart, et al.**

Dear Professor Olaf Eisen, dear Reviewers, dear *TC* readers,

We would like to thank both reviewers for very insightful and constructive reviews of our manuscript, as well as you and the editorial team for handling the review process.

We are very pleased to see that both reviewers recognised the importance of our results and how these were presented in our manuscript. Both reviewers have provided us with some excellent comments, which have undoubtedly improved the quality of our manuscript.

In the following response letter, we begin by addressing the comments from Reviewer #1, followed by those made by Reviewer #2. We have formatted the comments of each reviewer in italics, and have indented our responses in green below each comment. Please note that the line numbers provided in this response letter refer to the updated manuscript (non-tracked version), unless otherwise indicated.

Attached to this response letter are two versions of the manuscript, one with tracked changes ('Bodart_et_al_2023_resubmitted_tracked') and the final updated version with all changes incorporated but not highlighted ('Bodart_et_al_2023_resubmitted'). We have also provided the Supplementary Information documents, one with tracked changes ('Bodart_et_al_2023_supplement_resubmitted_tracked') and the final updated version with all changes incorporated but not highlighted ('Bodart_et_al_2023_supplement_resubmitted').

We look forward to hearing your decision and stand-by in the meantime with any queries you might have.

With best wishes,

Julien Bodart (on behalf of all co-authors)

**Reviewer #1:**

*General comments*

*In this work, the authors perform a widespread tracing of a ~4.72ka layer across significant region of West Antarctica. This is a valuable data set and the authors robustly show how it can be used with simple models to infer the average accumulation conditions around this time. The paper is clearly written and the outcomes clearly presented. I enjoyed reading this contribution and I have mostly minor comments to consider.*

> We would like to thank Reviewer #1, Michelle Koutnik, for her positive comments and useful suggestions, which have improved this manuscript.

*One potentially more major consideration, however, is that the lead statement in the abstract, "Modelling the past and future evolution of the West Antarctic Ice Sheet (WAIS) to atmospheric and ocean forcing is challenged by the availability and quality of observed paleo-boundary conditions", sets up this work for its value providing a new constraint on models. I agree that this is an important contribution, but more specific statements about how this new layer product and new inferences of past accumulation can be (or could be) used in models would be helpful. For example, it isn't clear to me if these constraints are valuable to regional ice-sheet models, continent-scale ice-sheet models, and/or also climate models. How could these outputs and inferences practically be used in models, and how many of the current generation of models are set up to use constraints like this in the way they have been shared? For example, few ice models assimilate layers, but is that the emphasis here – that even on Holocene timescales that layers are an important constraint on models, especially in the ice-sheet interior? Or, is it more to advocate using simple models to infer accumulation histories that can then more directly be used as boundary conditions in a wider suite of models? (Or, both!) If there are more specifics that can be added then I think it could have more impact on the modeling community. Some points are mentioned in the manuscript, but structuring parts of the manuscript around this more specifically would be worthwhile if it remains a main motivation and a main conclusion.*

> We thank Michelle Koutnik for her thorough review. We particularly appreciate her suggestions on how to more directly address the wider modelling community. We agree that the initial version of our manuscript did not sufficiently explore how the IRHs we use here, but also the inferences we make from them, may inform future ice-sheet modelling studies and how this can be effectively communicated in our paper.

> We believe that our results are useful for both types of modelling mentioned in the above comment. This includes using the conclusions drawn from our inferred accumulation rates as boundary conditions for modelling studies, but also integrating the information from the IRHs we use here as benchmarks for tuning ice-sheet model parameters (as we mention in the updated text). To answer this further, we do not believe that the IRHs we use would be suitable for informing atmospheric models intended to model Surface Mass Balance (SMB) across the ice sheet. This is because only shallow IRHs in the firn can recover information at the resolution required for annually resolving SMB, especially over the decadal time span that climate models usually cover.

> In order to answer the major consideration from Michelle Koutnik, we have re-written sections of the Abstract and Conclusion that provide much more specific details that will be of particular interest to the modelling community (see line-by-line comments below). In addition, we now provide a paragraph in the Introduction section (lines 121-138) and in the Discussion section (lines 463-479) which target specifically the modelling community. The

former provides an overview of how IRHs can be used in ice-sheet models, whereas the latter provides suggestions for future modelling studies based on our findings. Some finer details on these issues brought out by the reviewer's line-by-line comments are also outlined below in our response to those comments.

***Line-by-line comments***

*Line 25 (related to above point): If possible, I would suggest trying to clarify this point so it is more directly speaking to the modeling community. How do these results advance what is required for model spin up, and if these are continent-scale models then why is this region of West Antarctica and over this time so critical to improve model spin up? Minor rephrasing and a few more words could help make this a more impactful point*

We reworded this specific point accordingly, as follows:

"Spin-up of regional and continental ice-sheet models should include time-varying changes in Holocene accumulation rates from the WAIS Divide Ice Core to generate more realistic grounding-line evolution and past sea level rise contribution across this region".

*Line 38: Since mention comparison to modern in this sentence, could indicate time range over which 18% increase occurs during the mid-Holocene – would help clarify why increase of this amount is important (compared to seasonal or inter-annual variability, for example)*

Agreed and amended.

*Line 42-44: In general I feel like this point could be a bit more developed, especially if these results are meant to motivate ice-sheet modelers to use new records like this. There may not be space, but a few more words on why modeling past sea-level rise is important could be worthwhile. Elsewhere present and future sea level are also mentioned.*

We agree with this point. We therefore re-structured the start and end of our abstract, which we believe now provides more background that motivates ice-sheet modellers to consider our findings. The abstract now reads:

"Understanding the past and future evolution of the Antarctic Ice Sheet is challenged by the availability and quality of observed palaeo-boundary conditions. Numerical ice-sheet models often rely on these palaeo-boundary conditions to guide and evaluate their models' predictions of sea-level rise, with varying levels of confidence due to the sparsity of existing data across the ice sheet.

[…]

We find that our spatially-extensive, mid-Holocene-to-present accumulation estimates are consistent with a sustained late-Holocene period of higher accumulation rates occurring over millennia at the WAIS Divide Ice Core, thus highlighting the spatial representativeness of this ice core to the wider West Antarctic region. We conclude that future regional and continental ice-sheet modelling studies should base their climatic forcings on time-varying accumulation rates from the WAIS Divide Ice Core through the Holocene to generate more realistic predictions of the West Antarctic Ice Sheet's past contribution to sea-level rise."

*Line 54: Sentence is about modeling changes in ice volume and GL position, so wonder if some more recent references are worth including*

We added four recent additional references. Thank you.

*Line 56: This is a personal reaction, but the paper has a number of acronyms so it could be worth considering to keep only those that are necessary. I think "GL" could just be given as "grounding line", but again that may just be personal preference and I leave it to the authors to decide! [As an example, it became challenging to read in the text around Figure 1 – as many of these acronyms come into the text – but not sure the best way to handle that.]*

Thank you for this point. We agree that the words "grounding line" do not require an abbreviation and we have amended the text as a result. We have also removed the abbreviation "WD" (which stood for Western Divide in the original manuscript) as it is not necessary and also confusing with the other abbreviation used here for the WAIS Divide Ice Core; and have replaced it with "Amundsen-Weddell-Ross divide" throughout the updated paper. We believe that the other abbreviations are more useful and have thus decided not to amend them. Note that Figure 1 was missing key annotations and features due to an issue with the version of the figure which was provided in the original paper. The updated figure now provides these annotations.

*Also, could specify what grounding line you are referring to here*

Agreed and amended. The sentence now reads: "[…] parts of the grounding line of West Antarctica".

*Line 71: Could cut "At present"? Perhaps expand this point a bit to clarify if you mean climate models, ice-flow models, or both?*

Agreed and amended. The sentence now starts with: "Many numerical ice-sheet models that aim to predict [….]".

*Line 76: By "model physics" do you mean our physical understanding and how that can be represented in models? That could be through parameterization, but guessing that you are also indicating that some physics of the system are still unknown. Could be a bit more clear.*

We meant "model parameterisations". This sentence has been amended.

*Line 82 and 83: In this list of paleo records, are all of these available to inform WAIS? This paragraph could indicate what motivates the problem generally vs. specifically for West Antarctica. It would help to put in the context of whether the models are regional or continent-scale.*

Agreed. We have removed the mention "over the WAIS" in this sentence and widened our citations to other studies outside of the WAIS. The next paragraph now discusses more specifically the coverage of IRHs over the WAIS.

*Line 85: Could explain a bit more why radar is an "alternative data source"*

We have added a few words to explain why IRHs can be considered an alternative data source, as follows:

"A complimentary and spatially extensive alternative data source for inferring past climate across an ice sheet is provided by Internal Reflecting Horizons (IRHs) detected by RES. They primarily result from englacial acidity contrasts and are often detected horizontally for hundreds of kilometres on RES data (Harrison, 1973; Bingham and Siegert, 2007). When employed in combination with ice-core stratigraphies, IRHs can be used to extend age-depth

relationships away from an ice core by following peaks in electromagnetic power in the radar data (e.g. Beem et al., 2021; Bodart et al., 2021a; Cavitte et al., 2016; Jacobel and Welch, 2005; MacGregor et al., 2015; Whillans, 1976; Winter et al., 2019)."

*Line 92: It isn't completely clear what is meant by "suitable datasets" – entirely an issue with the radar data?*

    Agreed. We meant 'well-dated IRHs', which up until recently were lacking for this portion of Antarctica. We have replaced "suitable datasets" with "well-dated IRH datasets".

*Line 144: Since this is where the data sets are being introduced, could consider stating age range that they cover even more explicitly*

    Agreed and amended.

*Line 184-: Perhaps it is worth mentioning that these layers are well dated, which in this calculation of the relative uncertainty is what leads it to being a low estimate? Or, consider to bring the point up front in the paragraph about this not accounting for uncertainties in the model approximation. 3.3% uncertainty seems low so it could help the reader to explain that more and perhaps define more clearly what the relative uncertainty is*

    Thank you for this point. In response to Reviewer #2, we now provide an extensive analysis of accumulation uncertainties by assessing the structural uncertainties of the model used here (see Sections 2.2.2. and Supplementary Information). This results in much larger uncertainties (median: 14% of Nye-inferred accumulation rates across the grid) than when considering the IRH age uncertainty alone. This point is explored further in our response to Reviewer #2 below. The equation for the relative uncertainty is provided in Fig. S4.

*Line 194: I think that MacGregor et al. 2016 only used the LLA (but it would be worth double checking)*

    Agreed and amended.

*Line 200-: "...but because the value of D..." –I think that I understand the point being made, but I think it could be confusing so suggest revisiting*

    Agreed and amended.

*Line 205: This sentence could indicate that a vertical strain correction is still applied based on the model you apply ("sole result of accumulation rates at the surface" doesn't capture that)*

    Agreed and amended.

*Not sure if/where it belongs, but could also considering citing Leysinger Vieli et al. (2011), JGR Earth Surface 10.1029/2010JF001785 for their evaluation of the applicability of vertical flow models to infer accumulation rates from East Antarctic layers (though to find that a 3-D model was necessary)*

    Thank you. We already referenced this paper in Section 2.2.2 but added a mention at the top of Section 2.2.1 when discussing the use of multi-dimensional models.

*Could also consider citing Nielsen et al. (2015), Annals of Glaciology 56 (70), 70-78 as while they also find that a 2-D model is more appropriate they also compared a 1- D approach*

Agreed and added in Sect. 2.2.1. Thank you.

*Line 204: I think that MacGregor et al. 2009 applied a flowband model, not only the LLA so perhaps clarify why this is cited here; can see that Waddington et al. 2007 is cited in reference to the LLA itself (though they also applied a flowband model)*

We agree, although Waddington et al. (2007) and MacGregor et al. (2009) explain in detail where and why the LLA may be applicable, and where it may not be (which they then show with a flowband model). We have therefore kept these references there as we believe they are appropriate for this sentence.

*Line 212: "For this reason, the Nye model is generally only appropriate for IRHs found in the upper part of the ice column, as is the case here" – part of the value of using something like the LLA is that it can let you know where you may be surprised by relatively shallow layers (in the upper part of the ice column) but where the Nye model is not appropriate. This point made opens to this, but perhaps could be a chance to reiterate that it may need to be evaluated?*

Thank you for this point. We agree and now expand on this caveat in the methods section (particularly Sections 2.2.1-2.2.2) and in the Supplementary Information.

*Line 239: Is it necessary to clarify / explain that previous study cited (Medley et al., 2014) was inferring accumulation rates on different timescales and using firn layers – right?*

Agreed. We added: "which is comparable to previous studies that infer SMB from IRHs in the shallow firn (e.g., Medley et al., 2014)".

*Line 298: Can anything more be shared about why 100 km was picked?*

Agreed. We added: "when considering only the values that fall within 100 km of either side of the Amundsen-Weddell-Ross divide (i.e., in the accumulation zone of the Amundsen, Weddell, and Ross Sea sectors and where mean surface speeds average ~7 m a$^{-1}$)".

*Line 312: Are these numbers the same as ones given in lines 299? I got confused by how these two points were stated*

The numbers on Line 312 of the original manuscript referred to the comparison between the 4.72 ka-to-present accumulation rate grid and the 79 point-based accumulation measurements for modern times, as specified at the start of the paragraph. The number provided in Line 299 of the original manuscript was referring to the difference between the 4.72 ka-to-present accumulation rate grid and the modern accumulation rate grid from RACMO.

*Line 315: Perhaps indicate that point based are "cores", since that terminology comes up later. I had to read this sentence twice so perhaps stating even more explicitly (though it does make sense)*

Agreed and amended throughout.

*Line 338: Maybe use another phrase than "As a result…"*

Agreed and amended.

*Has RACMO2 modern or RACMO2 estimates for past decade(s) been evaluated against any ice-core data?*

> Yes, RACMO2 has been evaluated against >3,000 SMB in-situ measurements from shallow cores across Antarctica compiled by Favier et al. (2013) which we use here too. More details can be found in van Wessem et al. (2018) which we reference in our paper.

*Around Line 386: Is it worth commenting on what this means for recent signal of divide migration (Conway and Rasmussen, 2008)*

> Thank you for suggesting this addition. We added a few lines on the main conclusions of this paper in the Discussion section (lines 497-501).

*Line 399: The point "...a finding that must be considered by future modeling studies that simulate past sea-level rise from Antarctica since the LGM" could be strengthened to more directly speak to the modeling community (see above point)*

> This sentence was removed during the re-structuring of the Discussion. However, based on the suggestions from Michelle Koutnik, a new paragraph that provides suggestions for future modelling studies is now provided in the Discussion section (Sect. 4.1; lines 463-479). A condensed version is also included in the Abstract and Conclusion sections of our paper (see comments below). Also, note that a dedicated paragraph to modelling of IRHs is now included in the Introduction (lines 121-138).

*Around Line 415: The interior thickness response to an increase in accumulation in Koutnik et al. (2016) is also tied to assumptions about how fast the limited portion of the ice sheet interior in the model responds to changes in accumulation. This was incorporated in the model based on a physical assumption of ice-sheet adjustment, but it is still an assumption (could see Koutnik and Waddington, 2012). So, while the point made here is reasonable overall, I would suggest emphasizing that tens of meters adjustment depends on the model and also depends on the initial state – the adjustment through mid-Holocene may be sensitive to initial state (early Holocene) and accumulation history for previous few thousand years. And, good that you point out that dynamic component could be important and hasn't been directly taken into account.*

> Thank you for this point. We added a sentence on this on lines 490-493.

*So, the point made that "This potential increase in surface elevation is unlikely to affect the steady-state assumption of the 1-D model used here…" isn't necessarily founded on a suite of time-dependent flowband model runs that really tell you whether 10s of meters is the right range. One run gave this range, and bigger elevation change isn't necessarily expected, but this one model run doesn't really address the range of unknowns. Just a comment! Also, may want to clarify that you mean steady-state geometry.*

> Thank you for this point. We reformulated the start of this sentence to reflect this caveat.

*Line 418: I would suggest making a separate paragraph that is directed at the modeling community. This point is a good one, but stating more specifically how this could/should go would be helpful (I think)*

*Continent-scale modeling from LGM to present does estimate elevation change since the last glacial maximum but there are big discrepancies between models, and this problem isn't reconciled yet (though Jessica Badgeley who recently received her PhD at University of Washington is investigating this). All to say that I think not only mid-Holocene to present elevation change is important, but the potentially bigger changes considered from LGM to Holocene need to be reconciled between models and evaluated with available data (ice-core records, radar data). Is this important to address here – why the emphasis on capturing the mid-Holocene in models compared to the transition from LGM to*

*Holocene (I think ice-sheet adjustment times work as an explanation, but could consider if you need to address that).*

> Thank you. As stated above, we have now added a new paragraph in the Discussion section (in Section 4.1) which we believe targets more directly the modelling community. This is in addition to the additional paragraph we added in the Introduction which refers specifically to the use of IRHs in ice-sheet models, and in modifications brought to the Abstract and Conclusions in response to these comments.

*Lines 423-433: Is there more of a takeaway related to this work that should be made in relation to these points?*

> We believe that this has been explored well in the paper but perhaps in a confusing way since the two paragraphs (Lines 423-433 and Lines 434-457 in the original paper) addressing this point were separated in two instead of being one continuous paragraph. We have merged the two paragraphs in the updated version.

*Lines 434-457: This is interesting overall, but wonder if the framing in relation to this new work should be mentioned more up front in the paragraph. It was hard to put into context where this was going until about mid-paragraph. And, the rationale for connecting increased accumulation to interior thickening to GL advance seemed a bit tenuous given what was shared previously about not really knowing if there was thickening or thinning. The way I read this, there was a bit of back and forth in the discussion points between what may or may not be interpretable.*

*Would it be more impactful to consider structuring the discussion a bit more around how open questions may be addressed using available constraints, results from this work, and different types of models – what is really the path forward?*

*Is there a way to strengthen the point in lines 456-457, "potentially indicating that this sector is more controlled by changes in ice dynamics for which even moderate changes in accumulation rate cannot compensate" – in this case why would dynamic thinning not lead to a change in divide position, which may further complicate recovering an elevation change signal? Again, tying these points back into what is known and unknown may help.*

> Thank you for these points. We agree that this part of the Discussion was not always well structured and thus affected the main take-away messages from our paper. Based on these comments and those from Reviewer #2, we have re-structured these paragraphs to improve the connection with the rest of the Discussion. We have also added three sub-sections in the Discussion which frame the discussion around specific themes: (1) comparison with other Holocene accumulation estimates; (2) impact on ice-sheet elevation change during the Holocene; and (3) impact on grounding-line evolution during the Holocene. We hope this will significantly help with the readability and clarity of the Discussion section, alongside the other line-by-line changes we made throughout.

*Line 472: "making it a powerful dataset for ice-sheet models" – could be more explicit if you mean the layer data or the accumulation-rate inference? I think this is referring to the inference, but is this a data set?*

> This sentence has been removed from the conclusion based on recommendations from Reviewer #2. However note that a reformulated version of this sentence can now be found in the new modelling paragraph in Section 4.1 of the Discussion. This now reads:

"[…] In addition, we suggest that future modelling studies use the accumulation-rate variability from the WD14 Ice Core as a climate forcing in their ice-sheet models. Koutnik et al. (2016) previously showed that the WD14 record is unique in that it provides a reliable record of accumulation-rate variability during the Holocene, which other East Antarctic ice-core records often used to reconstruct the evolution of the WAIS do not possess. We found that these higher accumulation rates are spatially extensive across nearly one third of the WAIS, further suggesting that the WD14 Ice Core is indeed representative of the wider WAIS and can be used in regional or continental ice-sheet models as a reliable climate forcing for the region. Future regional and continental ice-sheet models should make use of this record to adjust their climatic boundary conditions to provide improved estimates of ice-elevation change and grounding-line evolution over Antarctica."

*How this becomes a powerful new constraint could be elaborated and clarified what classes of models would be the target. I am also not sure that the closing sentence connects clearly back to what is done here. What models use a "fixed Last Glacial Maximum value" (are these climate, ice, or both) and why does 4.72 ka constraint become so important from LGM to present – in practice what is being provided to (and suggested of) the modeling community? (see points above)*

Thank you for this suggestion. The sentence that contained the words "fixed Last Glacial Maximum value" has been removed based on comments from Reviewer #2 and as a result is not part of the updated version of our manuscript. As stated in our above responses, we now provide two comprehensive paragraphs (one in the Introduction, one in the Discussion) which target the modelling community directly. This is in addition to our restructuring and re-writing of the last part of the Conclusion section, which we believe is a significant improvement from our initial manuscript.
* * *
**Reviewer #2:**

*General comments*

*The authors present an analysis of a ubiquitous internal reflecting horizon across much of West Antarctica dated to 4.72 ka, which they use along with a simple 1-D ice flow model to infer a spatially varying, time-averaged accumulation rate for 4.72 ka to present. They compare this estimate with model- and ice core-based estimates for present-day accumulation rates and find that the 4.72ka–present accumulation rate was roughly 20% higher than more recent accumulation rates, with some interesting spatial details. Most notably, while the overall 4.72–present accumulation rates were higher than present day, the sites closest to the coast (at Thwaites and Pine Island glaciers) show the opposite pattern.*

*Overall I think this is a good manuscript and will be suitable for publication in The Cryosphere. I have listed a number of comments on details of the figures and text that should be addressed in the section below. The biggest weakness of the paper is the lack of an attempt to quantify uncertainty due to the very simple flow model that is used here. While assimilating these data into a 2- or 3-D model with higher-order dynamics is obviously far beyond the scope of this study, the Local Layer Approximation and the 1-D Nye model come with many assumptions and simplifications that I think are likely to yield larger uncertainties than the authors assume. However, the benefit of such a simple model is that it is presumably inexpensive and relatively fast to run. Thus, a metric of model uncertainty could potentially be calculated by sub-sampling the dataset and repeating the analysis many times. If formally quantifying model uncertainty is infeasible, it is important to at least demonstrate that the results are insensitive to the choice of upper bounds on D, normalized depth, and horizontal strain rate.*

*Waddington et al. (2007) define the LLA as being suitable when D << 1, rather than just D < 1 as is used here. Because of the imprecision of this definition, it is not clear that even D = 0.25 is sufficiently small for the LLA to hold. MacGregor et al. (2009) seem to think even D = 0.1 is too high to satisfy the D << 1 criterion: "Although we are using the three shallowest spatially extensive internal layers in the radar data, nearly all of our study area (96–98%) has D values >0.1 that do not satisfy the D << 1 criterion, suggesting that the LLA may not accurately infer accumulation rates." Their results also suggest that D might not actually be a good metric for determining where the LLA is suitable: "The values of D shown in Figures 3c and 4b suggest that the LLA is generally not suitable for flowband 1, but the small relative difference between $b_{LLA}$ and $b_{fb}$ for flowband 1 (5%) suggests that the LLA is acceptable." They also note that while the LLA does an acceptable job in some circumstances, a flow-band-based inverse model yields better results that differ significantly from the LLA in most cases. Thus, the validation of the LLA approach in this paper needs to be extended significantly beyond what is currently presented.*

*I am also skeptical that the LLA will be accurate in the lower regions of Thwaites and PIG within the model domain, where velocities exceed 300 m/yr, or in the regions where the 4.72 ka layer is deep in the ice column. It is also notable that the largest absolute values of Δb seem to occur in or near these areas. Care should be taken to show that there is not a significant correlation between Δb and D (or velocities or horizontal strain rates), and to evaluate whether the LLA applies when the layer is deep.*

We are thankful for the constructive review provided by Reviewer #2 which has helped us to improve the manuscript, particularly with regards to the assessment of uncertainties in the inferred accumulation rates.

Most of the comments from Reviewer #2 stem from the initial lack of uncertainties provided in the original version of our paper. In the updated version of our paper, we now provide an assessment of the structural uncertainty of the 1-D model and lay out more clearly the strengths and weaknesses of our approach. An updated subsection (Section 2.2.2) now focuses on these uncertainties, alongside new supplementary text and figures (Figures S3-S4). For context, our updated uncertainties now amount to 14% of the Nye-inferred accumulation rates over the whole grid, which is substantially greater than the previously reported value of <0.5% which considered only the age uncertainty of the IRH and not the effect of strain rates on the geometry of our IRHs. We believe that this assessment will satisfy the main concern of Reviewer #2 and refer them to the updated Methods section as well as Supplementary Information for more details.

With regards to the $D$ parameter, we agree that the notion of what is an 'acceptable' $D$ value is ambiguous. As Reviewer #2 mentioned, Waddington et al. (2007) and MacGregor et al. (2009) both state that the acceptability of the LLA relies on a value for $D$ that is within the bounds of $D \ll 1$. As Reviewer #2 also rightly mentioned in their review (quoted in line-by-line comments below), MacGregor et al. (2016) later accepted the notion that the LLA is acceptable for all areas where $D \leq 1$. In our paper, we followed MacGregor et al.'s later notion that all values falling within the boundary of $D \leq 1$ are acceptable, but with the caveat that ice-dynamical processes may disrupt the assumption that accumulation rates at the surface are the dominant cause for IRH depth in areas where this $D$ value is close to the $D = 1$ boundary. We accept that the original version of the paper did not explain this sufficiently and that this deserved more attention and clarity in our updated manuscript.

In the updated manuscript, we now use the statistical dispersion of the $D$ parameter ($D \leq 0.34$; amounting to 75% of the total D values over our grid) as a confidence metric, whereby all values situated within this boundary may satisfy the $D \ll 1$ criteria and those outside may require re-evaluating with the use of multi-dimensional models. To reflect these changes in the updated manuscript, we have updated the content of Sections 2.2.1-2.2.2 and updated Figure 4 and Figure S1.

More specific details on these actions are provided in our line-by-line comments below.

**Line-by-line comments**

*L 71: Important to note that only some modeling studies do this for the time-scales relevant here. For instance, studies using higher-order physics and high resolution cannot hope to use LGM-to-present reconstructions to calibrate and validate their models and instead rely on usually just a few years to decades of observations. And it's certainly fair to question whether lower-order ice sheet models like those used by the studies cited here should even be used to make predictions of WAIS changes over shorter (centennial) timescales.*

Thank you for this comment. We have re-worded this sentence and added the words 'long-term' to differentiate from century-scale sea level rise projections, as follows:

"Many numerical ice-sheet models that aim to predict Antarctica's long-term (past and future) contribution to […]"

*Figure 1 needs an inset context map showing location of this area within Antarctica*

Thank you for noticing this. We accidentally retained an incorrect early-draft figure in the original submission which lacked key features and annotations (including legend and inset map). We have updated this figure, which now includes these missing items.

*Section 2.2.1 and 2.2.2: Out of curiosity, could this model be used to place bounds on accumulation rates where D≥1?*

Thank you for your question. Technically, yes this model could be used to place bounds on accumulation rates in areas where $D$ is greater than unity; however as explained in the paper, the accuracy of the inferred-accumulation rates from the 1-D model depends on accumulation at the surface being a dominant factor in the shape and position of an IRH in the ice column, which is not a guaranteed criterion in areas where ice-dynamical processes (i.e. ice flow) play a larger role in influencing the geometry of IRHs.

*L 198: This is not correct. Waddington et al (2007) state that the LLA is valid only where D << 1, not where D ≤ 1 (see text below their eq 32). MacGregor et al. (2009) follow this definition. In contrast, MacGregor et al. (2016) do use D=1 as their threshold for the LLA, although I don't see where they explain this choice.*

Thank you for clarifying this. We agree with this point and accept this was an oversight from our part. We have changed this for the correct mathematical symbol where relevant throughout the paper.

An important consideration is that whilst MacGregor et al. (2016) do not explain why $D = 1$ is an acceptable threshold, Waddington et al. (2007) or McGregor et al. (2009) do not explain either what $D \ll 1$ should be, thus resulting in ambiguity around the actual threshold that is acceptable. We also note that one of the main conclusion from MacGregor et al. (2009) was that in the absence of numerous along-flow radar profiles necessary for constraining a flowband model (as is often the case over the ice sheet, including across our model domain), the LLA is the best way to infer accumulation rates from IRHs.

We have updated Section 2.2.1-2.2.2 to reflect this caveat and explain our reasoning for accepting the validity of the LLA for values smaller than unity, as follows:

"We quantified the effect of horizontal gradients on an IRH of age a by first estimating the total horizontal particle path length $L_{path}$ each "particle" of the 4.72 ka IRH has travelled since a, and then the characteristic lengths of variability in ice thickness ($L_H$) and apparent accumulation rate ($L_{\dot{b}}$) (Supplementary Information). These three components were then combined to generate a non-dimensional parameter D (Fig. S1d), which we used as a confidence metric for our inferred accumulation rates. Both Waddington et al. (2007) and MacGregor et al. (2009) suggested a value of D ≪ 1 over Antarctica, whereas MacGregor et al. (2016) used a maximum value of D = 1 to estimate where the LLA is acceptable over Greenland. Because D cannot be translated simply into an uncertainty in an LLA-inferred accumulation rate, it is not yet clear what exact value is appropriate. Smaller values of D indicate that local horizontal gradients in ice thickness and accumulation rates have a smaller effect on IRH depth of age $a$, and thus that the LLA may be valid (Waddington et al., 2007; MacGregor et al., 2009; 2016). Where D ≥ 1, the depth of an IRH is less likely to be the result of accumulation rates at the surface or vertical strain rates further down, and thus a more sophisticated model is likely required (Sect. 2.2.2) (Waddington et al., 2007). However, MacGregor et al. (2009) found that even along a flowband across Lake Vostok where the mean value of D is 0.50 for a 41-ka IRH, the difference in accumulation rate inferred from the LLA and from a more sophisticated flowband model could be relatively small (4-16%). This similarly suggests that accumulation rate can be inferred acceptably using the LLA where D is higher."

"For our study area, D values are mostly well below unity (median: 0.19; 25th quartile: 0.09; 75th quartile: 0.34), which suggests relatively little effect from ice-dynamical processes upon IRH depths across most of our grid. We used the upper quartile of the D distribution across our model domain (i.e., D ≤ 0.34) to show areas where we can have confidence that accumulation rate remains the dominant factor influencing the vertical position of our IRHs in the ice column (i.e., where the D ≪ 1 criterion is likely met; Fig. S1d). While accumulation rates inferred from IRHs situated in the upper quartile (Fig. S1d) may still be valid, we suggest additional caution in interpreting our results there due to the potential impact of larger flow gradients on IRH depths."

*Use of > and < on Figure S2 color bar limits are inconsistent*

Agreed and amended.

*Use of sigma for strain rates is unusual. Can you use epsilon_zz for vertical and epsilon_xx for longitudinal strain rates?*

Agreed and amended.

*Figure 2b shows that at some places the layer is ~80% of the ice depth. This goes against the assumptions in L 215–221: "where we can be reasonably confident that the ice sheet has remained close to steady-state and where IRHs are likely shallow enough not to have sustained appreciable disturbances that would affect the Nye model assumptions ". It doesn't look like the analysis is limited to depths shallower than 40% of the ice thickness. See other comments above and below about how the LLA needs more rigorous validation.*

Thank you for this comment. For clarity, we provide the median and interquartile range of our IRH depths in the paper (median: 40%; 25th quartile: 30%; 75th quartile: 50%; Fig. 2b-c). These show that for the most part, our IRHs are shallow enough for the LLA to be applicable, but we agree that a small amount of data points may be too deep in the ice column. However, we believe that the modifications made to Section 2.2, which now include both an assessment

of the uncertainties related to the model assumptions and more clarity on where we expect our inferred-accumulation rates to be less accurate with the use of the $D$ parameter as a confidence metric, help to answer this point.

*L223–226: But the analysis is extended to these areas of faster flow anyway, right? I don't see any evidence that these areas are left out of the analysis.*

Yes, we agree this is faster flow than at the divide, but our assessment of $L_{path}$ in Fig. S1a, and its contribution to the calculation of the $D$ parameter in Fig. S1d, suggest that this is still within the limits of the LLA if we accept the notion that the LLA is appropriate where $D \leq 1$.

In response to Reviewer #2's comments above, we now provide more clarity on where we expect the results from the LLA to be less accurate (see Sections 2.2.1-2.2.2 and Supplementary Information).

*Figure 3: Wording is a bit ambiguous in L306–307. Make it explicit that ice core accumulation rates are time-averaged for each core, not averaged across all 79 cores.*

Agreed and amended.

*Showing the difference w/ cores in Figures 3 and 4 is helpful, but a lot of the information is obscured where the cores are very close together. It might be helpful to visualize this as a scatter plot as well as a map view. It also looks like there are a fair number of cores that give the opposite sign to the data in the map, but it's not possible to tell from these figures how prevalent that is.*

Thank you for this suggestion. We now provide a scatter plot in the Supplementary Information (Fig. S6) and have added mention of this figure in the main text, where appropriate.

*Figure 3c: inconsistent use of > and < between top and bottom of this color bar*

Agreed and amended.

*L 342–345: This could also be an effect of the LLA being less appropriate at these lower elevations, where ice flow just happens to also be faster.*

Yes, we agree that this could be another reason (see response to comment below). This sentence has been modified since we now provide a catchment-by-catchment analysis of accumulation rates, as per Reviewer #2's suggestion below. However, we also added the caveat that the lack of a clear elevation-dependent gradient in accumulation rate over the Amundsen sector is likely related to the LLA being less accurate there (lines 405-411), as follow:

"We also note that whilst an elevation-dependent gradient in accumulation rates, dominated by high accumulation at the coast decreasing inland, exists over this sector for the mid-Holocene, it is much less marked than for present rates. This is not surprising, as this sector is where we observe the largest relative uncertainties in inferred accumulation rates across our grid (Fig. S4), indicating that the 1-D model is less able to produce realistic accumulation rates in the downstream end of our grid where ice flow is faster and strain rates are likely higher"

*Really interesting that 4.7ka–present accumulation rates on Thwaites and PIG are so much lower than modern! If that's real, presumably that's an elevation effect. It doesn't seem like this is discussed*

*in much detail, and you focus more on the WAIS Divide story. However, this is potentially a more interesting result than the result for WAIS Divide, so I'd encourage you to add more about this to the Discussion.*

Thank you for this point. As stated in the response above, we believe that the lower accumulation rates inferred from the Nye model at low elevations compared to the present are likely related to the LLA underestimating accumulation rates there due to the faster-flowing conditions in the downstream sections of our model domain.

We added a sentence in the Discussion section of our paper (Section 4.1) to encourage future modelling studies to use flowband models in this area to compare with our inferred accumulation rates (see lines 464-470), as follows:

"We suggest that future ice-sheet modelling studies investigate the difference in accumulation rates inferred from our 1-D model using multi-dimensional flowband models to assess effects of divergent and convergent flow on IRH depth and ultimately accumulation rates, as previously considered elsewhere in Antarctica (MacGregor et al., 2009). This could be conducted along a flowline transitioning from the slow-flowing regions directly downstream of the Amundsen-Weddell-Ross divide to the coastal margins of our grid, particularly over THW where we observe the largest uncertainties in accumulation rates."

*It seems like the biggest weakness of this study is the inability to quantify uncertainty due to the Nye model. Is there a way to estimate that? For instance, there might be areas within the domain where the Nye model actually doesn't work well, like in the tail of the normalized depth distribution in Fig 2c, or for other unknown reasons. Is there a way to re-do this analysis many times using sub-samples of the dataset to quantify the uncertainty due to the model? This could possibly be achieved by random sub-sampling, or by testing different thresholds of D or normalized Z. It would also be instructive to plot the inferred accumulation rate against D to ensure that D does not explain a significant amount of the variance in accumulation.*

Thank you for this comment. We believe that this has been answered in our responses to earlier comments. For more details on our new uncertainty assessment, please refer to Section 2.2.2 and the additional text and figures provided in the Supplementary Information.

*Can you report p-values for the comparison w/ RACMO as well as the comparison w/ ice cores?*

We have added these on lines 372-376.

*L 323–325: I don't understand the distinction between the two IRH-inferred accumulation rates in this sentence. Is the first one using a bilinear interpolation or some area average over the WD14 site?*

In response to the comments below, this sentence has been removed from the updated manuscript.

*Figure 5: As you allude to around L 347, I think it would be helpful to break this down into the IMIS and THW-PIG constituents. I think you should leave the current curves on the figure, but also add curves representing the same analysis for just IMIS and THW-PIG, respectively.*

Thank you for this suggestion. We agree and now provide a catchment-by-catchment analysis of accumulation rate and total accumulation rate for the Amundsen, Weddell and Ross sectors (Fig. 5).

*L 353: What does the "Fb" superscript here indicate?*

This was a typo. Thank you for noticing.

*L 373: This gives confidence that the Nye model works well at the ice divide, but that's where it is most likely to work given its assumptions. This does not necessarily mean that it works well across the whole domain, where flow starts to deviate from divide flow.*

We agree with this point. We have therefore removed the start of this paragraph in the updated manuscript.

*L 376: "This also suggests that the WD14 Ice Core suitably represents atmospheric conditions across the wider WD." How so? Can you explain further, and refer to a figure or table to help the reader understand the rationale?*

We agree that this point was not explained clearly. We meant that our finding, which provide a spatially-extensive record of higher accumulation rates across the Amundsen-Weddell-Ross divide matching a period of sustained increasing accumulation over millennial-scale at the Ice Core as shown by Fudge et al. (2016), suggests that the ice core is representative of the wider Amundsen-Weddell-Ross divide and can thus be used to model past changes over the ice sheet. Whilst this particular sentence has been removed from the updated manuscript, we now provide a clearer explanation for this throughout our paper, particularly at lines 435-442, as follows:

"These studies together point to a period of increasing accumulation observed at the WD14 Ice Core from ~7 ka onwards (Fudge et al., 2016; their Figure 2), with its peak matching the age of the 4.72 ka IRH used here. Thus, our accumulation-rate estimates likely form part of a wider pattern of a sustained increase in accumulation across the Amundsen-Weddell-Ross divide over several millennia. In showing that mean accumulation rates since 4.72 ka were 18% greater than modern rates modelled from RACMO2 across the Amundsen-Weddell-Ross divide, our results provide much wider regional support for the hypothesis that accumulation rates during the mid-Holocene exceeded modern rates across central West Antarctica."

*L 404: Your comparison is between mid- and late-Holocene/modern. There is nothing here to suggest that the accumulation rate at 4.72 ka was higher than >4.72 ka.*

Yes, thank you for spotting this inconsistency. We re-worded this sentence, as follows:

"[…] higher accumulation rates in the mid-Holocene relative to the present"

*L 420–422: Once again, I don't see how the work shown here validates the WD14 record as a proxy for West Antarctica, and this is the first place in which the temporal variability of the WD14 record is brought up. This needs to be re-evaluated or explained more clearly.*

We agree that this sentence was confusing and have rephrased it accordingly (see response to points above and below).

*L 445–450: Similar to comment ~L404: this argument is pointing the wrong way in time. Your inferred accumulation rates are higher than modern, but that does not imply that they represented an increase in accumulation rates relative to the earlier Holocene, which is what would be relevant for ice dynamics.*

Thank you for this comment. We agree that our wording was confusing, as per the above comments on this topic.

To avoid confusion, we rephrased the sentence in Lines 449-453 of the original manuscript and moved it higher up in this section to make clear that the higher accumulation rates we found here likely represent the peak of a sustained increase in accumulation rates recorded at the WAIS Divide Ice Core. Since we find that this is spatially extensive across most of the WAIS, this would have implications for ice sheet dynamics further downstream as recent studies suggest occurred across the Amundsen, Weddell and Ross sectors.

To improve clarity, and as suggested by Michelle Koutnik, we have also restructured this paragraph and divided the Discussion section into three sub-sections.

*L 455: There is a recent TCD preprint that does suggest moderate grounding line readvance in the late Holocene: https://tc.copernicus.org/preprints/tc-2022-172/*

Thank you for suggesting this new paper. We now discuss the main findings of this paper in lines 500-505.

*L 465: It would be almost impossible for there to be lower accumulation rates at the coast than at the interior in Antarctica in the absence of >0°C air temperatures.*

Thank you for this. We meant that the accumulation pattern over the divide is similar to today (i.e. higher accumulation on the Amundsen-side of the divide transitioning to lower accumulation on the Ross-side of the divide). This has been modified.

*L 471–472: Same issue as above. How do your results show this?*

This sentence has been removed from the updated manuscript and rephrased accordingly (see response to points above and below).

*The argument that ice sheet models need to account for time-varying accumulation rates seems like a straw man. Most ice sheet models do this in one way or another. While there are plenty of details about implementation that could be discussed, the point is not really brought up strongly elsewhere and feels out of place being stated so strongly in the conclusion. It has long been known that accumulation rates have changed from the LGM to present, and while the work presented here is a very useful and thorough addition to that story, it is not qualitatively different from what was known. While I agree that time-varying SMB should be taken into account, most models examining this period already do this. The difference between their various methods and the 18% difference between modern and average late-Holocene SMB is probably less important than model representation of ocean temperatures, sub-shelf melt rates, glacial isostatic adjustment, basal friction, low resolution and lower-order physics necessary for multi-millennium simulations, and other poorly understood processes that add enormous uncertainties to ice sheet modeling. It's certainly worth discussing, but I would recommend de-emphasizing this aspect as a primary conclusion of the paper, since no analysis regarding the impact of the change in accumulation rates on model results is presented here. What would be more helpful is to emphasize that since the WD14 record does seem to be representative of the region of interest, SMB forcing for models can potentially use a modern SMB spatial pattern (e.g., from RACMO) that is modulated over time according to the WD14 data.*

Thank you for this point. We agree that the way the sentence was written was confusing, and implied that this was one of the main finding of our paper. Instead, our intention was to provide suggestions for future modelling studies to use our inferred accumulation rates (or indeed those at the WAIS Divide Ice Core, as suggested here by Reviewer #2) to assess how the ice sheet would respond during the Holocene over the WAIS.

We have now rephrased this part of the Conclusion accordingly and provide more details in the re-structured Discussion (see response to earlier comments here).

*Supplement: paragraph above eq S1: Is "decimated" the correct word? What interpolation techniques were used to sample from native resolution to 1km, and from 1km back to 5km?*

Thank you. We have added some specifications in this sentence on what type of interpolation was used and replaced "decimated" with "subsampling".

*Fig S1 caption: Last sentence should say "where D<1", correct?*

Yes, thank you for spotting this typo.

---

## Editor Decision (ED1)

[revised manuscript text omitted]
 of accumulation with our those inferred from the 4.72 ka IRH-to-present estimates (see Sect.ion 2.4 for source references). The background colour map shows modern surface speeds from Rignot et al. (2017). Locations mentioned in this paper are abbreviated on the map, as follows: BYRD (Byrd Ice Core), IMIS (Institute and Möller Ice Streams), PIG (Pine Island Glacier), THW (Thwaites Glacier), WAIS (West Antarctic Ice Sheet), WD CD (Central Amundsen-Weddell-Ross DivideWestern Divide), WD14 (WAIS Divide Ice Core). Major ice divides are from Mouginot et al. (2017). The background image is the 2014 MODIS mosaic of Antarctica (Haran et al., 2018). For all analysis and figures in this study, the Projection for all figures in this paper is WGS84SCAR Antarctic Polar Stereographic projection is used (PSX/PSY; EPSG: 3031).

These RES surveys were used to track and date six IRHs spanning much of the Late Pleistocene and Holocene and Late Pleistocene (25.7 – 2.3 ka BP) that collectively covering much of the WAIS, across 
[revised manuscript text omitted]

**Supplementary Information**

**Assessing the suitability of the local-layer approximation**

To quantify to what extent the assumptions used in the 1-D model are valid for estimating Holocene accumulation rates between the 4.72 ka IRH and the present, we calculated horizontal gradients in modern ice thickness and accumulation rates over the WAIS, and combined these to calculate the non-dimensional parameter $D$ spanning the catchments where the 4.72 ka IRH was traced (Waddington et al., 2007) (Fig. S1).

The input datasets used for this calculation were modern ice thickness from BedMachine v2 (Morlighem, 2020), modern surface mass balance (1979 – 2019) from RACMO 2.3p2 (Van Wessem et al., 2018), and modern surface velocities (1996 – 2016) from the InSAR MEaSUREs v2 dataset (Rignot et al., 2017). These were all re-gridded to a single 1-km grid using bilinear interpolation and smoothed using an exponentially decaying filter equivalent to ten ice thicknesses in length, before subsampling the data to a common 5-km grid for data analysis. Following MacGregor et al. (2016), we re-calculated surface speed directions for slower ice-flow regions ($<100$ m a$^{-1}$) in the interior of the ice sheet using surface-elevation gradients from the BedMachine product. To calculate $L_{path}$ (Fig. S1a), we then produced a reverse flowline for each grid cell based on modern ice-surface velocity, $\bar{u}$, and calculated where along the reverse flowline we obtained age, $a$, as follows:

$$L_{path} = \bar{u}\,a . \tag{S1}$$

We then interpolated the ice-thickness and accumulation-rate grids onto each flowline and conducted a first-order polynomial fit to obtain the ice-thickness and accumulation gradients along the flowline. The ensuing gradients were then combined with the mean values along the flowline ($\bar{H}$ and $\bar{b}$) to calculate the characteristic lengths $L_H$ and $L_{\dot{b}}$ (Fig. S1b-c),

$$\frac{1}{L_H} = \left| \frac{1}{\bar{H}} \frac{dH}{dx} \right| . \tag{S2}$$

$$\frac{1}{L_{\dot{b}}} = \left| \frac{1}{\bar{\bar{b}}} \frac{d\dot{b}}{dx} \right| . \tag{S3}$$

Taken together, the ice-thickness and accumulation-rate gradients were combined to obtain a characteristic length scale, which was used to compare with $L_{path}$ to generate the non-dimensional parameter $D$ (Fig. S1d):

$$D = L_{path}\left(\frac{1}{L_H} + \frac{1}{L_b}\right). \tag{S4}$$

Values where $D \ll 1$ indicate that local horizontal gradients in ice thickness and accumulation rates have a smaller effect on IRH depth of age $a$, and hence we assume that the LLA is valid for estimating accumulation rates for an IRH of age $a$ (Waddington et al., 2007; MacGregor et al., 2009) (Sect. 2.2.1).

[Figure]

Figure S1. Suitability of the Local-Layer Approximation over the Pine Island, Thwaites, Institute and Möller ice-stream catchments for the 4.72 ka IRH. (a) Horizontal path length of a 4.72 ka particle of ice to reach its present location, calculated using modern surface velocities (Rignot et al., 2017). (b) Characteristic length of ice-thickness variability along the 4.72 ka particle path, estimated using modern ice thickness measurements from BedMachine v2 (Morlighem, 2020). (c) Characteristic length of accumulation variability along the 4.72 ka particle path, estimated using modern modelled surface mass balance data from RACMO2 (Van Wessem et al., 2018). (d) The $D$ parameter for the 4.72 ka IRH used to quantify the suitability of the LLA for the survey area. The white outline represents the model domain boundary used to model Holocene accumulation rates where $D$

≤ 1, whereas the black outline represents the upper limit of the interquartile range for the D parameter (i.e. $D \leq 0.34$) which we use to assess the level of confidence in the inferred Holocene accumulation rates.

**Estimating uncertainty in inferred accumulation rates**

Because the Nye model does not directly take into account the effect of strain rates on IRH depth and position within the ice column, it is not possible to assess its impact on the inferred accumulation rates, particularly in areas where strain rates are higher and the IRHs are deeper in the ice (e.g. the downstream section of our grid where ice flow is faster; Figs. 1-2 of the main paper). In turn, this limits our ability to quantify the model's structural uncertainty. Because the structural model uncertainty is likely larger than that related to the IRH age (Section 2.2 of the main paper), it is important to quantify it to assess the significance of accumulation-rate change from modern values that we detect.

To overcome this issue, we used the shallow-strain rate model developed by MacGregor et al. (2016) which includes a strain-rate parameter directly that is independent from ice thickness, rather than one that is tied to ice thickness as in the Nye model. The accumulation rates produced by this model are then used here to estimate lower and upper bounds in the accumulation rates that partly consider the effect of non-Nye vertical strain on the ice column and thus on the accumulation rate needed to reproduce the depth of the 4.72 ka IRH in the Nye model. The shallow-strain rate model is:

$$a(z) = \frac{1}{\dot{\epsilon}_{zz}^a} \ln \left( \frac{\dot{b}_a + \dot{\epsilon}_{zz}^a \, z_a}{\dot{b}_a} \right). \tag{Eq. S5}$$

The strain-rate parameter in Eq. (S5) would typically be $\dot{\epsilon}_{zz}^a$ from Figure S2a, but because this is calculated based on the results from Eq. (1) it is not independent from the inferred accumulation rates presented here and is thus not a suitable input for evaluating accumulation-rate uncertainty inferred from the Nye model. In the absence of well-constrained vertical strain rates across our grid, by continuity, we used the longitudinal strain rates ($\dot{\epsilon}_{xx}$; Fig. S2b) as an alternative to $\dot{\epsilon}_{zz}^a$ in the shallow-strain rate model (ignoring lateral strain).

These were calculated from gradients in the x and y-direction for modern surface speeds ($\bar{u}$) projected onto the appropriate surface velocity unit vectors ($\hat{u}_{||}$) (MacGregor et al., 2013):

$$\dot{\epsilon}_{xx} = \frac{\partial u}{\partial x} = \overline{\nabla} |\bar{u}| \cdot \hat{u}_{||}. \tag{Eq. S6}$$

[Figure]

Figure S2. Strain rate patterns across the survey area. (a) Mean Nye-inferred vertical strain rates, $\dot{\varepsilon}_{zz}^a$, for the 0-4.72 ka portion of the ice column calculated from Eq. (2) (b) Longitudinal strain rates,$\dot{\varepsilon}_{xx}$, obtained from Eq. (S6).

To assess whether $\dot{\varepsilon}_{xx}$ can be used as a proxy for $\dot{\varepsilon}_{zz}^a$, we solved Eq. S5 for $\dot{b}_a$, replaced $\dot{\varepsilon}_{zz}^a$ for $\dot{\varepsilon}_{xx}$ , and then compared the accumulation rate results inferred from the shallow-strain model to the Nye-inferred accumulation rates over our grid from Figure 3a. Note that $\dot{\varepsilon}_{xx}$ can only be used as a proxy for $\dot{\varepsilon}_{zz}^a$ where $\dot{\varepsilon}_{xx} > 0$, as positive $\dot{\varepsilon}_{zz}^a$ values are typically only found in areas where the ice column is expanding, such as the ablation zone, and are thus non-physical for our model domain. As a result, all negative $\dot{\varepsilon}_{xx}$ values were replaced by extremely low but positive strain-rate values ($10^{-7}$ $a^{-1}$). The results shown in Figure S3 demonstrate that both the Nye and shallow-strain models produce similar results, but with decreasing similarity where $D > 0.34$, which is likely related to ice-dynamical processes affecting the assumptions of the Nye model further downstream (Fig. S4).

[Figure]

Figure S3. Histograms of inferred accumulation rates from the Nye (a-d) and shallow-strain rates (e-h) models plotted against normalised IRH depths and binned into the four D quartiles (e.g. panels a and e are for all grid cells that fall within the lower quartile (Q1), b and f for all those that fall within the second quartile (Q2), etc; Sect. 2.2.1 of the main paper).

This analysis increases confidence that $\dot{\varepsilon}_{xx}$ can be used in the shallow-strain rate model from MacGregor et al. (2016) as a proxy for the vertical strain parameter, $\dot{\varepsilon}_{zz}^{a}$, to infer accumulation rates over the time period and location considered here, and thus ultimately has value for constraining uncertainty in the Nye-inferred accumulation rates (Fig. 3a). While this method likely produces more conservative uncertainty estimates than with the more challenging use of inverse flowband models that solves for the effect of changing flow, temperature and strain conditions along targeted flowbands, it enables a straightforward uncertainty quantification across a large area.

We then produced two sets of upper and lower accumulation-rate uncertainties ($\dot{b}_{4.72\,ka}^{Low}$ and $\dot{b}_{4.72\,ka}^{High}$) for each of the following products over our grid: (1) using the Nye model from Eq. 1 with the IRH age uncertainty (± 0.28 ka); and (2) same as (1) but using the shallow strain-rate model from Eq. S5 using $\dot{\varepsilon}_{xx}$ as a proxy for $\dot{\varepsilon}_{zz}^{a}$. We then calculated the maximum $\dot{b}_{4.72\,ka}^{Low}$ and $\dot{b}_{4.72\,ka}^{High}$ values for each grid cell (Fig. S4a-b) and combined these to provide a relative uncertainty to the Nye-inferred accumulation rates (Fig. S4c). The largest relative uncertainties to the Nye-inferred accumulation rates (> 70%) are found primarily across the downstream end of Thwaites Glacier, and to a smaller extent over the edges of the grid of Pine Island Glacier and Institute and Möller Ice Streams where longitudinal strain rates are higher due to faster flowing ice. Relatively low uncertainties are found across the Amundsen-Weddell-Ross divide and most of the region where $D \le 0.34$.

[Figure]

Figure S4. Uncertainties in inferred accumulation rates based on the radar and ice-core age uncertainties and from the accumulation rates returned from the shallow-strain rate model (Eq. S5). (a) Lower bound accumulation estimates, which are the product combination of the combined uncertainty from the radar and ice-core uncertainties in the age of the 4.72 ± 0.28 ka IRH (Muldoon et al., 2018; Bodart et al., 2021) and the accumulation rate returned from the shallow-strain rate model. (b) same as (a) but for the upper bounds in accumulation rates. (c) Relative uncertainty in Nye-inferred accumulation rates for the 4.72 ka IRH (Fig. 3a) based on the lower and upper bound estimates from Figures S4a-b.

[Figure]

Figure S5. Maximum distance to the nearest 500-m along-track point used for the interpolation of the 4.72 ka IRH depth and accumulation grids.

[Figure]

Figure S6. Scatter plot showing the difference in accumulation rates between the modern (cores and RACMO2) and the Holocene (4.72 ka) based on data showed in Figures 3c and 4 of the main paper. (a) Accumulation rates for each of Modern (cores), Modern (RACMO2), and Holocene (4.72 ka) at each of the 79 core locations shown in Figure 1. The five colour boxes at the top of (a) indicate the datasets to which each point belongs and are colour-coded as per the legend in Figure 1 (from left to right: MED14, ITASE, NEU08, SAMBA, SEAT-10). (b) Percentage change between Holocene and modern (cores; red) and Holocene and modern (RACMO2; blue) at the 79 core locations shown in Figure 1.

---

## Author Response (AR2)

Julien Bodart
School of GeoSciences
University of Edinburgh,
EH8 9XP, UK

Thursday, March 16th, 2023

**Re: [Paper ID #tc-2022-199] High mid-Holocene accumulation rates over West Antarctica inferred from a pervasive ice-penetrating radar reflector by J. A. Bodart, et al.**

Dear Professor Olaf Eisen,

Thank you very much for providing the additional minor comments and for accepting, in principle, our manuscript for submission to *The Cryosphere*. We provide a response to each minor comment in the below document, with your comments provided in italics and our response to them indented in green.

Attached to this response letter are two versions of the manuscript, one with tracked changes ('Bodart_et_al_2023_resubmitted_tracked') and the final updated version with all changes incorporated but not highlighted ('Bodart_et_al_2023_resubmitted'). We have also provided the Supplementary Information documents, one with tracked changes ('Bodart_et_al_2023_supplement_resubmitted_tracked') and the final updated version with all changes incorporated but not highlighted ('Bodart_et_al_2023_supplement_resubmitted').

With very best wishes,

Julien Bodart (on behalf of all co-authors)

**Editor comments**

*Dear Julien & coauthors,*

*thank you for your revised version of your manuscript and the final response to the reviews. Overall, I find that you considered the issues raised by the reviewers sufficiently adequate so that I can in principle accept your paper for publication in TC. Nevertheless, I would ask you to still make some final changes which I list below. Page and line numbers refer to the pdf with author track changes (ATC1 including SI), which I also attach together with the comments.*

*Looking forward to the final revision & thank you for submitting to TC.*

> Thank you very much for the additional comments and for accepting our paper. We particularly appreciate the attention to detail brought to our manuscript by yourself as the Editor.

*Minor comments*

*42: "When isochronal and dated at ice cores ..." Something is missing here, e.g. "are available"?*

> We have reformatted this sentence as follows: "When IRHs are isochronal and dated at ice cores, they can be used to determine palaeo-accumulation rates and patterns on large spatial scales"

*55: "at " -> "reconstructed from"*

> Agreed and amended

*56: highlighting: I find this is too strong, as criticized by reviewer 2. I suggest to use "indicating"*

> Agreed and amended

*116: "horizontally": I find this misleading, as they are almost never really horizontal, rephrase*

> The word "horizontally" has been removed.

*122: power -> return power*

> Agreed and amended

*155: shouldn't it be "to improve"?*

> Agreed and amended

*164: compare "them" - missing*

> Agreed and amended

*180: BBAS ???*

> "BBAS" was the name of the survey, as odd as this may sound. We are not sure what it stands for as there is no record of this that we could find. However, BAS refers to this survey consistently as "BBAS", so we have kept this as is in the paper.

*184: Should be 195 MHz, band 160–230 MHz, see Rodriguez-Morales 2014*

> Agreed and amended

*201: Polar Stereographic should be indicated as PS somewhere, as you deleted the previous explanation. Otherwise axis labels are not explained.*

> Agreed and amended

*217: 190 MHz - see above: 195*

> Agreed and amended

*228: model,*

> Agreed and amended

*235: epsilon_a Should be the same as in eqn. (2)*

> Agreed and amended

*241: Citation of BedMachine v2: the citation you provide is just one part according to the user guide ( https://nsidc.org/sites/default/files/nsidc-0756-v002-userguide_1.pdf ) which also asks users to do the following: "We also request that you acknowledge the author(s) of this data set by referencing the following peer-reviewed publication: Morlighem, M., E. Rignot, T. Binder, D. D. Blankenship, R. Drews, G. Eagles, O., et al. 2020. Deep glacial troughs and stabilizing ridges unveiled beneath the margins of the Antarctic ice sheet, Nature Geoscience. 13. 132-137. [https://doi.org/10.1038/s41561-019-0510-8](https://doi.org/10.1038/s41561-019-0510-8)" I therefore ask you to add this reference, as this is what open access data and FAIR is living of and requiring to be sustainable.*

> Strongly agree. This was added.

*243: provide a density here when introducing ice eq. for the first time for clarity. You do so only later in the manuscript.*

> Agreed and amended

*254/5: from ... are based - fix grammar*

> Agreed and amended. The sentence now reads "To quantify the suitability of the LLA which is used here to estimate our accumulation rate, […]."

*311: 1-2b-c: what does this refer to? Do you mean Fig 1, 2b, 2c?*

> Yes. This was amended. Thank you.

*351-355: Please split sentence, very long.*

> Agreed and amended

*452: add explanation of abbreviation for CD - central divide*

Agreed and amended

*472: You might want to add a short explanation on which side of the blue contour the values of D indicate more reliable results for the LLA, e.g. by referring to Fig. S1d.*

Agreed. We added "[…], whereby all values situated inside of this boundary may satisfy the $D \ll 1$ criteria and those outside may require re-evaluating with the use of multi-dimensional models […]."

*Code availability: consider the paper accepted, so you can publish*

Thank you. The codes and gridded depth + accumulation are now on Zenodo. The DOI is attached and the reference for the dataset is provided in the reference list.

*698: which timeline to you refer to with "in due course", i.e. when?*

Upon acceptance/publication; we are in the process of depositing the UTIG IRH on an open-access repository. We will update TC when the DOI is ready so that it can be included in the final version of the paper proofs.

*700? Frémand*

Agreed and amended

*704? consider it accepted and go ahead*

Agreed and amended (see reply to comment above)

*Few comments in the supplemental material which are not listed here, see pdf for details (pages attached at the end of the main text).*

Thank you. All additional comments in the Supplementary Material document were agreed and amended, as per your suggestions.